# Effects of nitrogen and phosphorus additions on nitrous oxide emission in a nitrogen-rich and two nitrogen-limited tropical forests

M. H. Zheng[1,5,†], T. Zhang[2,†], L. Liu[3], W. X. Zhu[4], W. Zhang[1], and J. M. Mo[1,*]

[1]Key Laboratory of Vegetation Restoration and Management of Degraded Ecosystems, South China Botanical Garden, Chinese Academy of Sciences, Guangzhou 510650, China

[2]School of Life Science and Technology, Lingnan Normal University, Zhanjiang 524048, China

[3]State Key Laboratory of Urban and Regional Ecology, Research Center for Eco-Environmental Science, Chinese Academy of Sciences, Beijing 100085, China

[4]Department of Biological Science, State University of New York – Binghamton, Binghamton NY, 13902, USA

[5]University of Chinese Academy of Sciences, Beijing 100039, China

[†] These authors contributed equally to this work.

*Correspondence to*: J. M. Mo (mojm@scib.ac.cn)

## Abstract

Nitrogen (N) deposition is generally considered to increase soil nitrous oxide ($N_2O$) emission in N-rich forests. In many tropical forests, however, elevated N deposition has caused soil N enrichment and further phosphorus (P) deficiency, and the interaction of N and P to control soil $N_2O$ emission remains poorly understood, particularly in forests with different soil N status. In this study, we examined the effects of N and P additions on soil $N_2O$ emission in an N-rich old-growth forest and two N-limited younger forests (a mixed and a pine forest) in southern China, to test the following hypotheses: (1) soil $N_2O$ emission is the highest in old-growth forest due to the N-rich soil; (2) N addition increases $N_2O$ emission more in the old-growth forest than in the two younger forests; (3) P addition decreases $N_2O$ emission more in the old-growth forest than in the two younger forests; and (4) P addition alleviates the stimulation of $N_2O$ emission by N addition. The following four treatments were established in each forest: Control, N addition (150 kg N ha$^{-1}$ yr$^{-1}$), P addition (150 kg P ha$^{-1}$ yr$^{-1}$), and NP addition (150 kg N ha$^{-1}$ yr$^{-1}$ plus 150 kg P ha$^{-1}$ yr$^{-1}$). From February 2007 to October 2009,

monthly quantification of soil $N_2O$ emission was performed using static chamber and gas chromatography techniques. Mean $N_2O$ emission was shown to be significantly higher in the old-growth forest ($13.9 \pm 0.7$ μg $N_2O$-N m$^{-2}$ h$^{-1}$) than in the mixed ($9.9 \pm 0.4$ μg $N_2O$-N m$^{-2}$ h$^{-1}$) or pine ($10.8 \pm 0.5$ μg $N_2O$-N m$^{-2}$ h$^{-1}$) forests, with no significant difference between the latter two. N addition significantly increased $N_2O$ emission in the old-growth forest but not in the two younger forests. However, both P- and NP-addition had no significant effect on $N_2O$ emission in all three forests, suggesting that P addition alleviated the stimulation of $N_2O$ emission by N addition in the old-growth forest. Although P fertilization may alleviate the stimulated effects of atmospheric N deposition on $N_2O$ emission in N-rich forests, this effect may only occur under high N deposition and/or long-term P addition, and we suggest future investigations to definitively assess this management strategy and the importance of P in regulating N cycles from regional to global scales.

## 1 Introduction

Nitrous oxide ($N_2O$) is a long-lived (atmospheric lifetime of approximately 114 years) greenhouse gas that has 298 times the ability of carbon dioxide ($CO_2$) to trap heat in the atmosphere (Cicerone, 1987; IPCC, 2007). It has been recognized as a major ozone-depleting substrate in the 21$^{st}$ century (Ravishankara et al., 2009). According to an estimation by the WMO (2012), atmospheric $N_2O$ concentration increased from 270 ppb during pre-industrial periods, to 324.2 ppb in 2011. The average emission rate of $N_2O$ increased by approximately 0.73−0.85 ppb yr$^{-1}$ from 1999 to 2005 (Hirsch et al., 2006; IPCC, 2007), and is predicted to continue increasing during the following decades (Bouwman et al., 2013). Global estimations show that soils, including agricultural soils and soils under natural vegetation, are dominant sources of atmospheric $N_2O$ (Hirsch et al., 2006; IPCC, 2007; Bouwman et al., 2013).

Tropical forest soils are important sources of $N_2O$, which is mainly produced by nitrification and denitrification (IPCC, 2007; Bouwman et al., 2013). At global scales, over half of the $N_2O$ emissions occur in the tropics (D'Amelio et al., 2009), of which tropical forests account for approximately 14−23% (IPCC, 2007). Because

tropical forest soils are often rich in N but poor in P, they are less able to retain external N input (Hall and Matson, 1999). With the greatest increases of atmospheric N deposition occurred in tropical regions (Galloway et al., 2008), tropical forests have shown a great increase in soil $N_2O$ emissions, compared with temperate and boreal forests (Matson and Vitousek, 1990). Although soil $N_2O$ emission is suggested to be regulated by soil

5   temperature, moisture, pH, and availability of nutrients (Werner et al., 2007; Rowlings et al., 2012), current knowledge on the factors controlling $N_2O$ emission in tropical forests is poor. This is not only because tropical forests have complicated structures and functions, as well as great temporal and spatial variations of $N_2O$ fluxes (D'Amelio et al., 2009; Zhu et al., 2013b), but also because only a small number of studies in tropical forests is available (Dalal and Allen, 2008; Liu and Greaver, 2009).

During recent decades, elevated atmospheric N deposition caused by anthropogenic activities has greatly altered terrestrial N cycles, reducing N input via biological N fixation and increasing N losses via $NO_3^-$ leaching and $N_2O$ emission (Vitousek et al., 1997b; Galloway et al., 2004). It is estimated that reactive N deposition increased from 34 Tg N $yr^{-1}$ in 1860, to 100 Tg N $yr^{-1}$ in 1995, and is expected to reach 200 Tg N $yr^{-1}$ by 2050

15   globally (Galloway et al., 2008). Tropical forests are often rich in N, and thus N deposition into such ecosystems will exceed their capacity for N retention (Aber et al., 1989), leading to rapid N losses via liquid leaching and gases emission (such as $N_2$, $N_2O$, NO, $NH_3$, and HONO). There are some field studies showing that N addition increased $N_2O$ emission in forests. For example, Hall and Matson (1999) reported significant increases in soil $N_2O$ emission after both short-term and long-term N addition in two Hawaiian forests. Zhang

20   et al. (2008b) suggested that N addition elevated soil $N_2O$ emission more readily in N-rich than N-limited forest. In a secondary tropical forest, Wang et al. (2014) also found a significant increase in $N_2O$ emission after 3 years of N fertilization. A meta-analysis by Liu and Greaver (2009) showed that N addition (10−562 kg N $ha^{-1}$ $yr^{-1}$) significantly increased $N_2O$ emission by approximately 216% across all ecosystems, among which tropical forests emitted the most.

In contrast to typically N-limited temperate forests, many tropical forests on highly weathered soils are rich in N but limited by phosphorus (P) (Vitousek and Matson, 1988; Vitousek et al., 2010). Hall and Matson (1999) reported that P-limited soils could emit more $N_2O$ than N-limited soils after N addition, suggesting an important role of P in controlling soil $N_2O$ emission. However, to date, studies on P-addition effects on soil $N_2O$ emission have mainly relied on incubation experiments (Sundareshwar et al., 2003; Mori et al., 2010, 2013; Baral et al., 2014), or have been limited to two tropical plantations (Mori et al., 2014; Zhang et al., 2014) and a secondary forest (Wang et al., 2014). Generally, these studies reported a decrease in soil $N_2O$ emission following P fertilization given the consequent increases in plant N uptake and/or microbial N immobilization, and thus reduced soil N availability for $N_2O$ production (Sundareshwar et al., 2003; Baral et al., 2014; Mori et al., 2014; Zhang et al., 2014). Only Mori et al. (2010, 2013) found a positive response of $N_2O$ emission to P addition, suggesting that P addition may stimulate soil N cycles and alleviate P limitation on nitrifying and denitrifying bacteria. Other than the studies above, similar work has not been carried out in other natural tropical forests. Moreover, in tropical forests with N-rich and P-limited conditions, the interaction of N and P to control soil $N_2O$ emission remains poorly understood (Hall and Matson, 2003; Wang et al., 2014).

We hypothesize that P addition may reduce soil $N_2O$ emission in tropical forests based on two lines of evidences. First, in several P-limited tropical forests or plantations, P addition significantly increased root N uptake capacity (Treseder and Vitousek, 2001) and aboveground plant N contents (Fernandez et al., 2000; Pampolina et al., 2002; Graciano et al., 2006). Second, our previous study found that P addition significantly increased soil microbial communities (Liu et al., 2012) and marginally increased microbial biomass N (Liu et al., 2013) in an N-rich tropical forest. Such findings indicate the potential capacity of P to increase N uptake and immobilization, thus decreasing N losses in tropical forests. However, the capacity of P to reduce N losses may be related to forest development. Despite many tropical forests have rich N in soils, several younger forests early in soil development are still N-limited (Vitousek et al., 1997a). Compared with the old-growth forests, younger forests often show the higher N demands and utilization of plants and microbes, but the lower

rates of soil N cycling, such as mineralization, nitrification and leaching (Aber et al., 1998). In contrast, old-growth forests have the higher P demand because they are commonly depleted in P (Vitousek et al., 2010). For example, a previous study showed that soil microbes and/or tree roots released more phosphatase in an old-growth forest than in the younger one (Zheng et al., 2015). Based on these evidence and considering current

knowledge gaps regarding nutrient (N and P) control of $N_2O$ emission in tropical forests, we conducted a randomized factorial design experiment to investigate the effects of N and P addition on soil $N_2O$ emission in three tropical forests in southern China: an N-rich old-growth forest, and two N-limited younger forests (a mixed and a pine forest). We hypothesized that: (1) soil $N_2O$ emission is the highest in old-growth forest due to the N-rich soil; (2) N addition increases $N_2O$ emission more in the old-growth forest than in the two younger

forests; (3) P addition decreases $N_2O$ emission more in the old-growth forest than in the two younger forests; and (4) P addition alleviates the stimulation of $N_2O$ emission by N addition.

## 2 Materials and Methods

### 2.1 Site description

This study was conducted in the Dinghushan Biosphere Reserve (DHSBR), located in the center of Guangdong Province, southern China (112º10' E, 23º10' N). The reserve occupies an area of approximately 1200 ha and includes three forests: an old-growth forest and two younger forests (a mixed broadleaf/pine forest and a pine forest). The old-growth forest has been well protected from human disturbance for over 400 years, with major species such as *Castanopsis chinensis* Hance, *Schima superba* Chardn. & Champ., *Cryptocarya chinensis* (Hance)

Hemsl., *Cryptocarya concinna* Hance, *Machilus chinensis* (Champ. Ex Benth.) Hemsl., and *Syzygium rehderianum* Merr. & Perry in the tree layer and *Calamus rhabdicladus* Burret, *Ardisia quinquegona* Bl., and *Hemigramma decurrens* (Hook.) Copel. in the understory layer (Wang et al., 1982). The two younger forests both originated from a 1930s clear-cut and subsequent pine plantation establishment (Mo et al., 2006, 2007b). They experienced continuous human disturbance (the harvesting of understory and litter) from 1930 to 1956

(mixed forest) and 1998 (pine forest). Because of the colonization from natural dispersal of regional broadleaf

species, the mixed forest contains both pine- and broadleaf-tree species (Mo et al., 2003, 2007b). The mixed forest is dominated by *Pinus* (P) *massoniana*, *Schima superba* Chardn. & Champ., *Castanopsis chinensis* Hance, *Craibiodendron kwangtungense* S. Y. Hu, *Lindera metcalfiana* Allen, and *Cryptocarya concinna* Hance, while the pine forest is dominated by *P. massoniana*.

Earlier studies suggested that $22-28$ kg N ha$^{-1}$yr$^{-1}$ were retained in the upper 20cm soil and the plant biomass (including canopy trees, understory plants and forest litter) in the two younger forests, and that a net loss of $8-16$ kg N ha$^{-1}$yr$^{-1}$ mainly via dissolve inorganic N ($NH_4^+$ and $NO_3^-$) leaching and soil $N_2O$ emission occurred in the old-growth forest (Fang et al., 2008). This indicates N saturation in the old-growth forest but N limitation in

10    the two younger ones. Different soil N status is also supported by different litter decomposition rates, with negative N effects in the old-growth forest but positive effects in the two younger forests (Mo et al., 2006). The N-rich status of the old-growth forest is also directly supported by its higher foliar N:P ratios ($20.6-36.8$) compared with the two younger forests (13.8 in pine forest and $17.8-24.4$ in mixed forest) (Huang et al., 2013). However, soil P is deficient in the old-growth forest, as evidenced by the positive responses of soil $CH_4$ uptake

15    (Zhang et al., 2011), microbial biomass (Liu et al., 2012) and live fine root biomass (Zhu et al., 2013a) to P addition.

The reserve has a typical humid monsoon climate with an average annual precipitation of 1927 mm, about 75% of which falls from March to August and only 6% from December to February as reported by our previous

20    studies (Huang and Fan, 1982; Lu et al., 2008). The mean annual temperature is 21 ºC with a January mean temperature of 12.6 ºC and July mean temperature of 28.0 ºC; annual mean relative humidity is 80% (Huang and Fan, 1982). Wet inorganic N deposition was 34, 24, and 26 kg N ha$^{-1}$yr$^{-1}$ in 2004 and 2005 for the old-growth, mixed, and pine forests, respectively, with an additional input of $15-20$ kg N ha$^{-1}$yr$^{-1}$ as dissolved organic N (Fang et al., 2008). All forest soils are lateritic red earth formed from sandstone, and soil depth is < 30cm,

30−60cm, and >60cm in the old-growth, mixed, and pine forests, respectively (Mo et al., 2003). General soil properties are listed in Table 1.

## 2.2 Experimental design

The experiment was established in 2007 with five replicates of each four treatments in each forest: Control (no fertilization), N addition (150 kg N ha$^{-1}$ yr$^{-1}$), P addition (150 kg P ha$^{-1}$ yr$^{-1}$), and NP addition (150 kg N ha$^{-1}$ yr$^{-1}$ plus 150 kg P ha$^{-1}$ yr$^{-1}$),with a total of 20 plots (5 m × 5 m). Each plot was surrounded by a 5m wide buffer strip. We used the high N gradient, about 3 folds of atmospheric N deposition rate, because many soil processes responded significantly only under this gradient in the old-growth forest (Mo et al., 2008; Zhang et al., 2008a; Lu et al., 2010). High P gradient was used because of the high P demand of soil microbes in the old-growth forest (Liu et al., 2012). Although the two younger forests are N-limited, we used the similar N and P gradients for the main purpose of comparison among the forests (Zheng et al., 2015; Zhu et al., 2013a). High fertilization rates can remove all possible N and P constraints in both young and old-growth forests (Cleveland and Townsend, 2006). In addition, plot size and fertilizer level in our forests were also the same as those in Costa Rica by Cleveland and Townsend (2006). All plots and treatments were assigned randomly. NH$_4$NO$_3$ and NaH$_2$PO$_4$ solutions were used as fertilizers and sprayed below the canopy using a backpack sprayer, bimonthly from February 2007 to October 2009. Fertilizer was weighed and mixed with 5L of water for each plot. Each control plot received 5L of water without fertilizer.

## 2.3 N$_2$O flux measurement

N$_2$O fluxes were measured from January 2007 before the first fertilizer application. Two static chambers were installed in each plot in November 2006, two months prior to the gas sampling. The chamber design and measurement method were adopted from Zhang et al. (2011). Gas fluxes were monitored monthly using a static chamber and a gas chromatograph (Agilent 4890D). Each static chamber consisted of an anchor ring and a removable cover chamber. The anchor ring was a PVC pipe (25 cm diameter and 16 cm height) permanently

anchored into the soil to 8 cm depth. During gas collection, a removable cover chamber (25 cm diameter and 30 cm height) was attached tightly to the anchor ring using a rubber O-ring seal. Gas samples were collected from each chamber from 9:00−10:00 at local time, during which the greenhouse gas fluxes are closer to the daily means (Tang et al., 2006). Gas samples were taken with a 60-ml plastic syringe at 0, 15, and 30 min intervals after the chamber closure. Before each sampling, syringes were flushed three times with chamber gas to mix the headspace. The gases samples were analyzed within 12 h on the gas chromatograph (Agilent 4890D) fitted with an electron capture detector (ECD) for $N_2O$. Two stainless steel columns (pre-column and main-column was 1m and 3m in length, respectively) packed with Porapak Q were used to separate $N_2O$. The oven temperature and ECD temperature was 55 ºC and 330 ºC, respectively. To avoid the interference of $CO_2$ from the gas samples which can lead to overestimation of $N_2O$ fluxes as suggested by Zheng et al. (2008), we used $N_2$ as the carrier gas (flow rate of 35mL min$^{-1}$) and introduced 10% of $CO_2$ in $N_2$ as the make-up gas (flow rate of 2mL min$^{-1}$) into the ECD (Wang et al., 2010). Through introducing high concentration and low flow rate of $CO_2$ into the ECD, the interference of $CO_2$ from the gas samples is negligible (Wang et al., 2010). Calibration gases ($N_2O$ at 321 ppbv, bottle's No. 070811) were obtained from the Institute of Atmospheric Physics, Chinese Academy of Sciences.

The calculation of $N_2O$ fluxes followed the method of Holland et al. (1999), based on linear regression of chamber gas concentration with time. Atmospheric pressure was measured at the sampling sites using an air pressure gauge (Model THOMMEN 2000, Switzerland). Meanwhile, air temperature (enclosure), soil temperature (at 5 cm depth) and moisture (0−10 cm depth) inside each chamber, were measured during each sampling. Soil moisture content was detected using a TDR-probe (Model Top TZS-I, China), and converted to water filled pore space (WFPS) according to the following formula:

WFPS = Vol / (1-SBD / 2.65)

SBD: soil bulk density (g cm$^{-3}$); Vol: volumetric water moisture (%); 2.65 g cm$^{-3}$ is the assumed particle density in mineral soil of forests (Linn et al., 1984). It is possible that the particle density value may be different between

forest types (old-growth vs. younger forests), but we focused on the comparison between treatments in this study, so this case is of minor importance.

## 2.4 Soil sample analyses

Soil sampling was conducted in February 2007 (before the first fertilizer application), August 2007, February 2008, August 2008, February 2009, and August 2009. Five soil cores (2.5 cm inner diameter) were collected randomly from 0−10 cm soil depths and mixed by plot. Soil pH was measured in a soil/water (1:2.5) suspension. Soil organic carbon (C) was measured using dichromate oxidation and titration with ferrous ammonium sulfate (Liu, 1996). Soil microbial biomass C was measured using the chloroform fumigation-extraction method

(Vance et al., 1987). Soil dissolved organic C was extracted with 0.5 M $K_2SO_4$ and analyzed using a total carbon analyzer (Shimadzu model TOC-500, Kyoto, Japan). Total N concentration was measured using semimicro-Kjeldahl digestion followed by detection of ammonium on a Wescan ammonia analyzer, and total P concentration was measured spectrophotometrically after acidified ammonium persulfate digestion (Anderson and Ingram, 1989). Soil available P was measured spectrophotometrically after extraction with acid-ammonium

fluoride solution (Liu, 1996). Soil $NH_4^+$-N was measured spectrophotometrically by the indophenol blue method after extraction with potassium chloride solution (Liu, 1996).

Soil nitrification rate was measured according to the *in situ* incubation method described by Raison et al. (1987). Briefly, 10 soil cores (2.5 cm inner diameter) were collected from each plot, 5 of which were brought to the

laboratory for measurement of soil $NO_3^-$−N using cadmium reduction followed by sulfanilamide-NAD reaction, and the remainders were returned to the plots for 1 month incubation. Nitrification rate was calculated from the difference between extractable $NO_3^-$−N contents before and after incubation.

## 2.5 Statistical analyses

Repeated measures analysis of variance was used to examine the effect of fertilizer treatments on soil $N_2O$

emission and soil properties from February 2007 to October 2009. Two-way ANOVA was used to determine the treatment effects on soil $N_2O$ emission. One-way ANOVA was used to determine the differences in soil properties among treatments for each sampling. Linear regression analyses were used to determine the relationships between $N_2O$ emission and soil WFPS / soil temperature in each forest. Data were tested for normality (Kolmogorov-Smirnov test) and equality (Levene's test) of variances, and were log-transformed for analysis if they did not meet the requirements of normality or equality of variances. All analyses were conducted using the SPSS 16.0 for windows (SPSS Inc., Chicago, IL, USA). Statistically significant differences were recognized at $P < 0.05$, unless otherwise stated.

## 3 Results

### 3.1 Soil temperature

Soil temperature (at 5 cm depth) showed a similar pattern in all plots across the three forests, increasing from spring to summer and decreasing from fall to winter (Fig. 1). The mean soil temperature of the control plots during the study period was $21.8 \pm 0.4$, $22.6 \pm 0.4$, and $23.4 \pm 0.4$ °C in the old-growth, mixed, and pine forests, respectively. Repeated measures ANOVA highlighted significant differences ($P < 0.001$) in soil temperatures between each forest. In the mixed forest, soil temperature was significantly lower in P-addition plots ($P = 0.043$) compared to the control plots, while N- and NP-addition had no effect on soil temperature. No treatment effect was detected on soil temperature in the old-growth and pine forests, as determined by repeated measures ANOVA.

### 3.2 Soil WFPS

Soil WFPS (0−10 cm depth) increased in all forests from dry winter to wet spring, but decreased in summer, possibly due to the higher plant uptake and transpiration, despite the high amount of precipitation in summer (Fig. 2). Mean soil WFPS in control plots during the study period was $31.1 \pm 1.1$, $29.5 \pm 1.2$, and $28.3 \pm 1.2$ % in the old-growth, mixed, and pine forests, respectively. Repeated measures ANOVA showed no significant

difference of soil WFPS in the control plots among three forests. N-, P-, and NP-addition had no significant effect on soil WFPS in any forest, as determined by repeated measures ANOVA.

### 3.3 Soil properties

Repeated measures ANOVA showed that soil pH significantly increased after P-addition in the old-growth forest (Table 2). Soil $NO_3^-$ concentrations significantly decreased after P-addition in the old-growth and mixed forests, and significantly increased after N-addition in the pine forest. Soil $NH_4^+$ concentrations and total inorganic N ($NH_4^+ + NO_3^-$) concentrations had no response to either N- or P-addition in any forest. Soil available P concentrations significantly increased after P-addition in all the forests. Soil organic C significantly increased after N-addition in the mixed and pine forests, but not in the old-growth forest. Soil microbial biomass C significantly increased after P-addition in the old-growth forest and after N-addition in the mixed forest. Interaction of combined N and P additions occurred in soil AP concentrations and microbial biomass C in the old-growth forest, and in soil pH and $NO_3^-$ concentrations in the mixed forest.

### 3.4 Soil $N_2O$ emission in control plots

Soil $N_2O$ emission was higher in all forests during spring and summer, and lower in fall and winter (Fig. 3). Mean soil $N_2O$ emission was $14.0 \pm 0.7$, $9.9 \pm 0.4$, and $10.9 \pm 0.5$ µg $N_2O$-N $m^{-2}$ $h^{-1}$ in the old-growth, mixed, and pine forests, respectively (Fig. 4), with being significantly higher in the old-growth forest than in the mixed ($P = 0.001$) and pine ($P = 0.005$) forests. In the control plots, soil temperature and WFPS showed a significant positive linear relationship with soil $N_2O$ emission (Fig. 5), and explained 9−17% and 12−23% of $N_2O$ fluxes variation across the forests (Table 4). The models that included soil temperature and WFPS as parameters showed the higher $R^2$ values (22−28%; Table 4).

### 3.5 Soil $N_2O$ emission after N and P addition

Effects of N- and P-addition on soil $N_2O$ emission varied with forest type (Fig. 4). In the old-growth forest,

mean $N_2O$ emission during the study period was 24.7% higher in the N-addition plots (17.4 ± 1.1 µg $N_2O$-N $m^{-2}$ $h^{-1}$), not significantly different in the P-addition plots (14.0 ± 0.8 µg $N_2O$-N $m^{-2}$ $h^{-1}$), and 13.9% higher in the NP-addition plots (15.9 ± 0.9 µg $N_2O$-N $m^{-2}$ $h^{-1}$), compared to the control plots (14.0 ± 0.7 µg $N_2O$-N $m^{-2}$ $h^{-1}$). However, significant differences were confined to the N-addition treatment ($P$ = 0.036). In the mixed forest, mean $N_2O$ emission slightly increased by 0.7, 8.0 and 3.9% after N-, P-, and NP-addition, respectively. In the pine forest, N- and NP-addition slightly increased mean $N_2O$ emission by 1.1 and 14.7%, respectively, while P-addition marginally decreased mean $N_2O$ emission by 2.5%. In the mixed and pine forest, no significant differences among treatments were identified by repeated measures ANOVA.

Two-way ANOVA highlighted the significant positive effects of N-addition on $N_2O$ emission in spring 2007, fall 2007, winter 2008 and fall 2008, and the marginal negative effects of P-addition in fall 2008 and summer 2009, in the old-growth forest (Table 3). In contrast, only a significant positive effect of N-addition occurred in winter 2008 in the mixed forest, and in spring 2007, fall 2008 in the pine forest. Interactive effects ($P$ < 0.1) of combined N and P additions occurred in the old-growth (winter 2008), mixed (fall 2007, winter 2008 and winter 2009), and pine (summer 2007, winter 2009) forests.

### 3.6 Soil nitrification rate

In the old-growth forest, N-addition significantly increased soil nitrification rate ($P$ = 0.005), while P- and NP-addition had no significant effect (Fig. 6). In the mixed and pine forest, soil nitrification rate was not affected by N- or/and P-addition.

### 4 Discussion

### 4.1 $N_2O$ emission in control plots

Soil $N_2O$ emissions measured in the present study (9.9−13.9 µg $N_2O$-N $m^{-2}$ $h^{-1}$) were comparable to previous reports from tropical forests (10.0−11.5 µg $N_2O$-N $m^{-2}$ $h^{-1}$) (Kiese et al., 2008; Neto et al., 2011). However, our

results were lower than those from adjacent forests (24.1−69.0 µg $N_2O$-N $m^{-2}$ $h^{-1}$) (Tang et al., 2006; Zhang et al., 2008b) and other tropical forests (16.3−77.1 µg $N_2O$-N $m^{-2}$ $h^{-1}$) (Kiese et al., 2008; Davidson et al., 2008; Konda et al., 2010), and higher than those from many tropical/subtropical forests (1.0−8.7 µg $N_2O$-N $m^{-2}$ $h^{-1}$) (Hall et al., 2004; Werner et al., 2006; Wang et al., 2010; Wieder et al., 2011). Taken together, these data

suggest a high variation in $N_2O$ emission among different study regions, possibly due to the difference in soil types and/or climatic conditions.

As expected, a generally higher seasonal $N_2O$ emission and a significantly higher mean $N_2O$ emission were identified in the old-growth forest than in the two younger forests (Fig. 3 and 4), suggesting that $N_2O$ emission may vary depending on forest type. $N_2O$ emission has been suggested to increases with succession (Verchot et

al., 1999; Erickson et al., 2001), possibly due to the increase in soil N content (Erickson et al., 2002). For example, soil N enrichment due to the presence of N-fixing legume trees has been linked with higher $N_2O$ emission (Erickson et al., 2002; Arai et al., 2008; Konda et al., 2010; Zhang et al., 2014). In addition, higher $N_2O$ emission in N-rich soils has been reported by a study in adjacent forests with different soil N status (Zhang

et al., 2008b). These findings are consistent with our results in that the old-growth forest had higher inorganic N ($NH_4^+$ and $NO_3^-$) and total N content than the mixed and pine forests (Table 1). Given almost complete saturation of N in this old-growth forest, investigated previously by Fang et al (2008), excess N in soils would be readily lost as dissolved organic and inorganic N (Fang et al., 2008, 2009), and $N_2O$ gas (Zhang et al., 2008b). Thus, our results further confirm that N-rich forests have a higher $N_2O$ emission than N-limited forests.

In addition to soil N status, soil pH and the availability of other nutrients may account for higher $N_2O$ emission in the old-growth forest. Compared to the younger forests, the old-growth forest had more acid soil conditions (Table 1 and 2), likely supporting the higher chemo-denitrification (Tate, 1995; Chalk and Smith, 1983; Mørkved et al., 2007). Additionally, the old-growth forest had significantly higher soil dissolved organic C and

total organic C (Table 1), which could provide more C energy for $N_2O$ production (Zhang et al., 2014). N-rich

and P-limiting conditions have also been suggested to support higher $N_2O$ emission (Zhang et al., 2008b). In the present study, soil N:P ratios were significantly higher in the old-growth forest than in the mixed and pine forest (Table 1), suggesting that low availability of soil P may intensify $N_2O$ emission under N-rich conditions (Zhang et al., 2014), thus indicating the potential interaction of N and P to control $N_2O$ emission.

### 4.2 Effects of soil temperature and WFPS on $N_2O$ emission

Overall, soil temperature increased from spring to summer but decreased from fall to winter in all the forest plots (Fig. 1). $N_2O$ emission was positively correlated to soil temperature in all three forests (Table 4 and Fig. 5), which was consistent with previous studies in tropical forests (Butterbach-Bahl et al., 2004; Zhang et al., 2008b; 10 Zhu et al., 2013b; Zhang et al., 2014). However, mean soil temperature was highest in the pine forest, followed by the mixed and old-growth forests (statistical difference of $P < 0.001$ between each forest), which was inconsistent with the patterns of mean $N_2O$ emission identified across forests (with being significantly higher in the old-growth forest than in the mixed ($P = 0.001$) and pine ($P = 0.005$) forests; Fig. 4). This suggests a limited ability of soil temperature to explain the pattern in $N_2O$ emission across forests with different soil N status.

Compared to the models with soil temperature and $N_2O$ fluxes as parameters, the $R^2$ values of the models with soil WFPS and $N_2O$ fluxes as parameters were not much higher (Table 4). However, mean soil WFPS showed comparable dynamics to mean $N_2O$ emission, with the highest WFPS in the old-growth forest and the lowest WFPS in the pine forest (Fig. 2). In each forest, soil WFPS showed a positive relationship with $N_2O$ emission 20 (Fig. 5), as has previously been observed across forests with different soil N status (Zhang et al., 2008b, 2014). Moreover, seasonal patterns in soil WFPS (Fig. 2) and $N_2O$ emission were comparable in all forests (Fig. 3), suggesting that soil WFPS can predict the seasonal variance of $N_2O$ emission, as follows. In spring, forest soil was enriched with inorganic N (accumulated during non-growing seasons mainly due to the lack of rainfall) (Mo et al., 2003) and had higher WFPS (increased in wet seasons); conditions that would generate a pulsing 25 effect, because wetting dry soil will trigger emissions of $N_2O$ and other nitrogenous gases by stimulating

microbial consumption of soil $NH_4^+$ and/or $NO_3^-$ (Davidson et al., 2000; Butterbach-Bahl et al., 2004; Werner et al., 2006). In summer, $N_2O$ emission began to decrease given decreasing soil WPFS (Fig. 3) possibly caused by the higher plant uptake and transpiration (Cheng et al., 2015). In fall and winter, both the lower soil inorganic N (decreased after growing seasons) (Mo et al., 2003) and WPFS (decreased in dry seasons) suppressed $N_2O$

production. Accordingly, $N_2O$ emission was highest in spring, declined in summer, and was lowest in fall and winter (Fig. 3). Thus, our findings suggest that soil WFPS may be a more appropriate predictor of $N_2O$ emission in forests with different soil N status than soil temperature.

### 4.3 Effects of N addition on $N_2O$ emission

As expected, N addition significantly increased mean $N_2O$ emission in the old-growth forest, but not in the mixed and pine forests (Fig. 4), which was consistent with the results from adjacent forests (Zhang et al., 2008b). In several N-rich forests, $N_2O$ emission significantly increased after N addition (Hall and Matson, 1999; Venterea et al., 2003; Koehler et al., 2009; Zhang et al., 2014), whereas it was hardly impacted by N input in the N-limited forests (Davidson et al., 2000; Skiba et al., 2004), or only increased after chronic N addition

(Magill et al., 2000; Hall and Matson, 2003). This indicates an important control of $N_2O$ emission by soil N status (Zhang et al., 2008b), as explained below.

As supported by our results, additional N inputs to N-rich forests exceed the ecosystems capacity for N retention, and thus less N is utilized (Aber et al., 1998). In the old-growth forest, we found no increase in soil

organic C, microbial biomass C (Table 2), or litter decomposition rate (Mo et al., 2006) after N addition, whereas live fine root biomass was shown to decrease (Zhu et al., 2013a), suggesting that N addition no longer increases soil and plant C pools in this forest. Moreover, N fertilizer application rate was much larger than atmospheric N deposition rate, leading to excess soil N accumulating in the old-growth forest which would favor nitrifying and denitrifying bacteria (Zhang et al., 2008b), and therefore significantly stimulated soil

nitrification rate (Fig. 6), $N_2O$ emission (Fig. 4) and $NO_3^-$ leaching (Fang et al., 2009). As a result, no

significant increase in soil inorganic N ($NH_4^+$ and $NO_3^-$) was observed after N addition in the old-growth forest (Table 2). Thus, in combination with previous findings, our results confirm that N addition will increase $N_2O$ emission in N-rich forests.

In contrast, in N-limited forests, N is retained to support plant and microbial growth, and/or accumulation of soil organic matter (Aber et al., 1998; Harrington et al., 2001). In the mixed and pine forests, two N-limited ecosystems (Mo et al., 2006), despite no significant increase in soil total inorganic N following N addition, a significant increase in soil microbial biomass C and soil organic C was observed in the mixed forest, as well as a significant increase in soil organic C in the pine forest (Table 2). Both forests showed positive responses of

litter decomposition rate to N addition (Mo et al., 2006), but no net N losses via $NO_3^-$ leaching (Fang et al., 2008). In addition, our previous study showed that under atmospheric N deposition, the N retention in the two forests was in accordance with the estimates of N accumulation in plant biomass and litter increment (Mo et al., 2004, 2007a; Fang et al., 2008), suggesting that the N retention was mainly used for plant growth rather than gaseous N loss. In this study, despite we did not measure other gases losses (such as $NH_3$, NO, HONO and $NO_2$)

which are also important in forest soils, we found that $N_2O$ emission showed no response to N addition in either forest (Fig. 4), and nitrification rate did not change (Fig. 6). Although rates of N addition in the present study were much higher than atmospheric N deposition, all above evidences suggest that N continues to be utilized rather than $N_2O$ emission following N addition in our N-limited forests. Further studies are needed to examine whether N addition increases other nitrogenous gases loss in the N-limited forests.

### 4.4 Effects of P addition on $N_2O$ emission

No significant change in mean $N_2O$ emission was observed following P addition in any of the study forests (Fig. 4), allowing us to reject the hypothesis that P addition causes greater decrease in $N_2O$ emission in the old-growth forest than in two younger forests. This finding was inconsistent with many previous studies

conducted in situ (Mori et al., 2014; Zhang et al., 2014) or in laboratories (Sundareshwar et al., 2003; Mori et

al., 2010, 2013; Baral et al., 2014). For example, Mori et al. (2014) and Zhang et al. (2014) reported that P addition significantly decreased $N_2O$ emission in a leguminous and non-leguminous plantation, respectively. Under laboratory conditions, Sundareshwar et al. (2003) found a negative response of sediment $N_2O$ emission to phosphate addition. Based on a pot experiment with maize, Baral et al. (2014) also suggested that alleviation of P limitation would decrease $N_2O$ emission. The major mechanism of this P-driven decrease in $N_2O$ emission is the increased plant uptake of soil N due to higher P availability, which therefore reduces N availability for nitrifying and denitrifying bacteria (Mori et al., 2010). However, several incubation experiments found a positive response of $N_2O$ emission to P addition (Mori et al., 2010, 2013), with authors suggesting that P addition might stimulate soil N cycles for nitrification and denitrification and/or might alleviate soil P limitation of nitrifying and denitrifying bacteria. In contrast, a lack of response of $N_2O$ emission to P addition has rarely been reported, especially for natural forests (Wang et al., 2014), and the mechanism remains poorly understood.

Based on the present study, we propose that a lack of response of $N_2O$ emission to P addition may be attributed to failure of soil N immobilization, or N uptake stimulated by short-term P addition. P fertilization has been suggested to decrease soil N substrates (or increase soil N immobilization), and thus suppress $N_2O$ production (Sundareshwar et al., 2003; Mori et al., 2010, 2014; Zhang et al., 2014). However, we found no significant change in soil total inorganic N ($NH_4^+$ plus $NO_3^-$) after P addition in all forests, despite a significant decrease in $NO_3^-$ in the old-growth and mixed forests (Table 2). Moreover, soil nitrification rate remained stable after P addition in all forests (Fig. 6), suggesting that P addition did not affect $N_2O$ production in the present study. Yet, in a recent study, significant decreases in soil inorganic N and $N_2O$ emission occurred after 6 years of P addition in an old-growth forest (Chen et al., 2015), indicating that $N_2O$ emission may remain stable following short-term P addition, but decrease after long-term addition in N-rich forests. We further suggest studies to identify whether long-term P addition will also decrease $N_2O$ emission in N-limited forests.

**4.5 Effects of combined N and P additions on $N_2O$ emission**

Consistent with our hypothesis, mean $N_2O$ emission showed no response to combined N and P additions in all forests (Fig. 4), suggesting that P alleviated the stimulating effect of N addition on $N_2O$ emission in the old-growth forest; as has been reported by several previous studies. For example, Hall and Matson (2003) reported that N addition significantly increased soil $N_2O$ emission but N and P addition had no effect in a

P-limited forest. Using a pot experiment, Baral et al. (2014) found that $N_2O$ emission was highest under N fertilization treatment, but reduced after P fertilization in a P-limited soil/sand mixture. Zhang et al. (2014) also reported that $N_2O$ emission significantly increased with N addition but not with NP addition in a leguminous plantation. However, our results were inconsistent with those of Mori et al. (2013) and Wang et al. (2014), who suggested that both N- and NP-addition significantly increased $N_2O$ emission.

Currently, two mechanisms of the P alleviation of $N_2O$ emission are plausible. First, P addition may alleviate P limitation of plants, and thus increase plant uptake of N (Hall and Matson, 1999; Baral et al., 2014; Sundareshwar et al., 2003). Second, P addition may alleviate P limitation of soil microbes and therefore increase microbial N immobilization (Sundareshwar et al., 2003). Both pathways will reduce soil N substrates

available for $N_2O$ production. Although plant and microbial N contents were not measured in this study, our recent study in the old-growth forest found no effect of 5 years of P- and NP-addition on fine root N contents (Zhu et al., 2013a). P addition likely alleviated the P limitation on soil microbes in our old-growth forest, because our previous study showed that P addition significantly increased soil microbial biomass and soil respiration (Liu et al., 2012). Compared with the controls, P addition changed soil microbial community,

including the increases in biomass of bacteria and AM fungi (Liu et al., 2012, 2013). The increases in AM fungi may help plants acquire more N and P nutrients (Tresede and Vitousek, 2001), because they are more efficient in obtaining nutrients from the soil than the plant roots (Liu et al., 2013). In addition, the increases in bacterial and fungal biomass may potentially increase total N acquirement, as evidenced by our previous study showing that 4 years of P- and NP-addition tended to increase soil microbial biomass N (Liu et al., 2013). Accordingly, P

alleviation of the N stimulation on $N_2O$ emission in our old-growth forest was likely attributed to an increase in

microbial N immobilization. NP addition did not significantly affect soil total inorganic N ($NH_4^+$ plus $NO_3^-$) (Table S2), and thus soil nitrification rate (Fig. 6), which in turn did not affect $N_2O$ emission.

It is interesting that soil $N_2O$ emission reduced after P addition compared with that after N addition (150 kg N ha$^{-1}$ yr$^{-1}$), but not when compared with that under atmospheric N deposition (~50 kg N ha$^{-1}$ yr$^{-1}$). We infer this may be related to the levels of N addition and/or the period of P addition. First, it is possible that low N addition, such as atmospheric N deposition in our study, may not cause a significant increase in soil $N_2O$ emission in this N-rich forest. Our previous study showed that under atmospheric N deposition (49.5 kg N ha$^{-1}$ yr$^{-1}$), soil had higher N leaching (59.8 kg N ha$^{-1}$ yr$^{-1}$) in this N-rich forest, suggesting a net N loss under atmospheric N deposition (low N input), and thus the less N retained for $N_2O$ production (Fang et al., 2008). Accordingly, it is possible that low N addition fail to increase soil $N_2O$ emission in the N-rich forest, and thus P addition may show no alleviated effect. Second, a lack of response of $N_2O$ emission to P addition compared with the control may also be related to the P fertilization period. Nutrients (N and P) addition in our study was only applied for about 2 years, and we did not observe the alleviated effect of P addition on soil $N_2O$ emission under atmospheric N deposition (Fig. 3 and 4). However, our recent study in the same forest found that long-term (6 years) P addition significantly decreased soil $N_2O$ emission compared with the control (atmospheric N deposition) (Chen et al., 2015). This suggests that fertilization period is also an important factor affecting the alleviated effect of P addition on $N_2O$ emission in this N-rich forest. Therefore, our findings suggest that P addition will alleviate the stimulating effects of N on $N_2O$ emission in the N-rich forest, but this effect may only occur under high N addition and/or long-term P addition.

**5 Conclusions**

To our knowledge, this is the first study to examine how N and P interact to control soil $N_2O$ emission in tropical forests with different soil N status. Our results confirm that N-rich forests have higher $N_2O$ emission than N-limited forests, and N addition will merely increase $N_2O$ emission in N-rich forests, as less N is utilized

in N-rich soils. However, neither P- nor NP-addition affects $N_2O$ emission in both N-rich and N-limited forests, which suggests that P addition potentially alleviates N stimulation of $N_2O$ emission in N-rich forests, with the potential mechanism of microbial N immobilization, but this alleviated effect may only occur under high N addition and/or long-term P addition. Therefore, P fertilization can be used to reduce soil $N_2O$ emission in N-rich forests under atmospheric N deposition, but we suggest more investigations to definitively assess this management strategy and the importance of P in regulating N cycles from regional to global scales.

**Acknowledgements**

This study was financially supported by the National Natural Science Foundation of China (NO: 41273143 and 41473112), the Natural Science of Guangdong Province (2014A030311023) and the Research Found for the Doctoral Program of Lingnan Normal University (ZL 1202). We thank four anonymous referees for their comments and suggestions on the manuscript.

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

**Table 1.** General characteristics of the 0−10cm mineral soils in the three study forests.

| Forest type | Old-growth forest | Mixed forest | Pine forest |
|---|---|---|---|
| pH value ($H_2O$) | 3.9(0.0) b | 4.0(0.0) a | 4.0(0.1) ab |
| $NH_4^+$ (mg kg$^{-1}$) | 2.4(0.3) a | 1.4(0.1) b | 2.4(0.0) a |
| $NO_3^-$ (mg kg$^{-1}$) | 4.3(0.3) a | 1.3(0.2) c | 3.3(0.5) b |
| Dissolve organic C (mg kg$^{-1}$) | 709.2(33.7) a | 552.3(13.9) b | 573.2(25.2) b |
| Soil organic C (%) | 4.1(0.2) a | 2.8(0.2) b | 2.9(0.3) b |
| Microbial biomass C (mg kg$^{-1}$) | 551.9(38.5) a | 75.9(7.0) c | 165.6(10.3) b |
| Available P (mg kg$^{-1}$) | 2.1(0.4) a | 0.9(0.1) b | 1.1(0.2) b |
| Total N (g kg$^{-1}$) | 1.6(0.1) a | 1.1(0.2) b | 1.1(0.1) b |
| Total P (mg g$^{-1}$) | 0.5(0.0) | 0.5(0.0) | 0.5(0.0) |
| N:P ratios | 3.2(0.2) a | 2.2(0.2) b | 2.3(0.3) b |

Notes: Soil samples were collected in February 2007. Values are means with standard error in parentheses (n = 5). Different lowercase letters indicate significant differences among forests, as determined by one-way ANOVA ($P <$ 0.05).

**Table 2.** Effects of N and P addition on soil properties in the three study forests.

| Statistical analyses | | | | | | Repeated measures ANOVA | | |
|---|---|---|---|---|---|---|---|---|
| Treatment | | C | N | P | NP | N | P | N×P |
| Old-growth forest | pH | 3.9(0.0) | 3.9(0.0) | 4.0(0.0) | 3.9(0.0) | ns | ** | ns |
| | $NH_4^+$ (mg kg$^{-1}$) | 8.7(0.8) | 10.1(0.9) | 9.7(0.8) | 10.2(1.2) | ns | ns | ns |
| | $NO_3^-$ (mg kg$^{-1}$) | 5.9(0.6) | 6.6(0.8) | 4.5(0.8) | 3.5(0.5) | ns | ** | ns |
| | $NH_4^+ + NO_3^-$ (mg kg$^{-1}$) | 14.6(1.1) | 16.7(1.5) | 14.2(1.2) | 13.8(1.4) | ns | ns | ns |
| | Available P (mg kg$^{-1}$) | 1.4(0.2) | 2.7(0.6) | 8.7(1.4) | 5.8(1.0) | ns | ** | * |
| | SOC (%) | 4.1(0.2) | 4.6(0.2) | 4.8(0.1) | 4.4(0.2) | ns | ns | ns |
| | MBC (mg kg$^{-1}$) | 434.3(42.7) | 359.9(41.2) | 422.6(44.9) | 488.6(60.0) | ns | * | * |
| Mixed forest | pH | 4.0(0.0) | 4.1(0.0) | 4.1(0.0) | 4.0(0.0) | ns | ns | * |
| | $NH_4^+$ (mg kg$^{-1}$) | 8.4(0.9) | 8.9(0.9) | 8.9(0.9) | 9.0(0.9) | ns | ns | ns |
| | $NO_3^-$ (mg kg$^{-1}$) | 1.9(0.3) | 1.8(0.3) | 1.2(0.2) | 1.8(0.3) | ns | * | ** |
| | $NH_4^+ + NO_3^-$ (mg kg$^{-1}$) | 10.3(1.1) | 10.7(1.0) | 10.1(1.0) | 10.8(1.0) | ns | ns | ns |
| | Available P (mg kg$^{-1}$) | 1.4(0.2) | 3.7(0.8) | 7.2(1.3) | 5.7(1.1) | ns | ** | ns |
| | SOC (%) | 2.4(0.1) | 2.8(0.1) | 2.7(0.1) | 2.9(0.1) | * | ns | ns |
| | MBC (mg kg$^{-1}$) | 239.7(23.6) | 254.5(25.69) | 240.5(31.9) | 291.5(31.8) | * | ns | ns |
| Pine forest | pH | 4.0(0.0) | 4.0(0.0) | 4.0(0.0) | 4.0(0.0) | ns | ns | ns |
| | $NH_4^+$ (mg kg$^{-1}$) | 8.8(0.7) | 8.5(1.0) | 8.3(0.9) | 8.9(1.1) | ns | ns | ns |
| | $NO_3^-$ (mg kg$^{-1}$) | 2.9(0.4) | 3.2(0.4) | 2.3(0.4) | 3.2(0.4) | * | ns | ns |
| | $NH_4^+ + NO_3^-$ (mg kg$^{-1}$) | 11.7(1.0) | 11.7(1.2) | 10.6(1.1) | 12.1(1.2) | ns | ns | ns |
| | Available P (mg kg$^{-1}$) | 1.9(0.4) | 1.5(0.4) | 7.7(1.4) | 7.2(1.4) | ns | ** | ns |
| | SOC (%) | 3.3(0.1) | 3.6(0.2) | 3.4(0.1) | 3.9(0.2) | * | ns | ns |
| | MBC (mg kg$^{-1}$) | 306.2(41.1) | 274.3(40.7) | 260.2(33.1) | 268.2(34.7) | ns | ns | ns |

Notes: Soils were sampled in August 2007, February 2008, August 2008, February 2009, and August 2009. February and August is within the dry and wet season, respectively in the study region. Values shown in Table 2 are means (all the sampling periods) with standard error in parentheses (n = 25), and the values of each sampling period are shown in Table S2−S4 in the supporting information. SOC: soil organic carbon; MBC: microbial biomass carbon. '**', '*' and 'ns' represent statistical difference of $P<0.01$, $P<0.05$ and $P>0.05$, respectively.

**Table 3.** $P$ value of two-way repeated measures ANOVA of seasonal $N_2O$ fluxes in the three study forests.

| Seasons | Spring 2007 | Summer 2007 | Fall 2007 | Winter 2008 | Spring 2008 | Summer 2008 | Fall 2008 | Winter 2009 | Spring 2009 | Summer 2009 |
|---------|-------------|-------------|-----------|-------------|-------------|-------------|-----------|-------------|-------------|-------------|
| Old-growth forest | | | | | | | | | | |
| N | **0.001** | 0.470 | **0.048** | **0.021** | 0.631 | 0.761 | **0.029** | 0.253 | 0.567 | 0.775 |
| P | 0.328 | 0.519 | 0.552 | 0.265 | 0.383 | 0.931 | **0.090** | 0.356 | 0.524 | **0.052** |
| N×P | 0.531 | 0.748 | 0.556 | **0.034** | 0.751 | 0.519 | 0.782 | 0.565 | 0.202 | 0.172 |
| Mixed forest | | | | | | | | | | |
| N | 0.881 | 0.667 | 0.253 | **0.017** | 0.304 | 0.866 | 0.609 | 0.446 | 0.989 | 0.349 |
| P | 0.601 | 0.948 | 0.462 | 0.128 | 0.522 | 0.649 | 0.570 | 0.958 | 0.277 | 0.102 |
| N×P | 0.721 | 0.487 | **0.084** | **0.043** | 0.814 | 0.440 | 0.470 | **0.089** | 0.509 | 0.711 |
| Pine forest | | | | | | | | | | |
| N | **0.027** | 0.101 | 0.934 | 0.255 | 0.612 | 0.793 | **0.045** | 0.907 | 0.762 | 0.651 |
| P | 0.559 | 0.117 | 0.152 | 0.600 | 0.743 | 0.875 | 0.898 | 0.234 | 0.912 | 0.410 |
| N×P | 0.491 | **0.024** | 0.163 | 0.431 | 0.685 | 0.194 | 0.400 | **0.097** | 0.834 | 0.434 |

Notes: Spring from April to June, summer from July to September, fall from October to December and winter from January to March. $P$ values that are less than 0.1 are marked by bold type.

**Table 4.** Models for the relationships between $N_2O$ fluxes ($N_2O$), soil WFPS (W) and temperature (T) in the control plots of the study forests.

| Parameters | Forest type | Regression models | *P* value | $R^2$ | n |
|---|---|---|---|---|---|
| $N_2O$, W | Old-growth forest | $N_2O=0.22\times W+6.80$ | <0.001 | 0.12 | 169 |
| | Mixed forest | $N_2O=0.13\times W+5.77$ | <0.001 | 0.18 | 168 |
| | Pine forest | $N_2O=0.20\times W+5.15$ | <0.001 | 0.23 | 168 |
| $N_2O$, T | Old-growth forest | $N_2O=0.79\times T-3.61$ | <0.001 | 0.17 | 169 |
| | Mixed forest | $N_2O=0.32\times T+2.45$ | <0.001 | 0.11 | 168 |
| | Pine forest | $N_2O=0.39\times T+1.58$ | <0.001 | 0.09 | 169 |
| $N_2O$, W, T | Old-growth forest | $N_2O=0.64\times T+0.14\times W-4.75$ | <0.001 | 0.22 | 169 |
| | Mixed forest | $N_2O=0.22\times T+0.11\times W+1.53$ | <0.001 | 0.22 | 168 |
| | Pine forest | $N_2O=0.28\times T+0.18\times W-0.86$ | <0.001 | 0.28 | 168 |

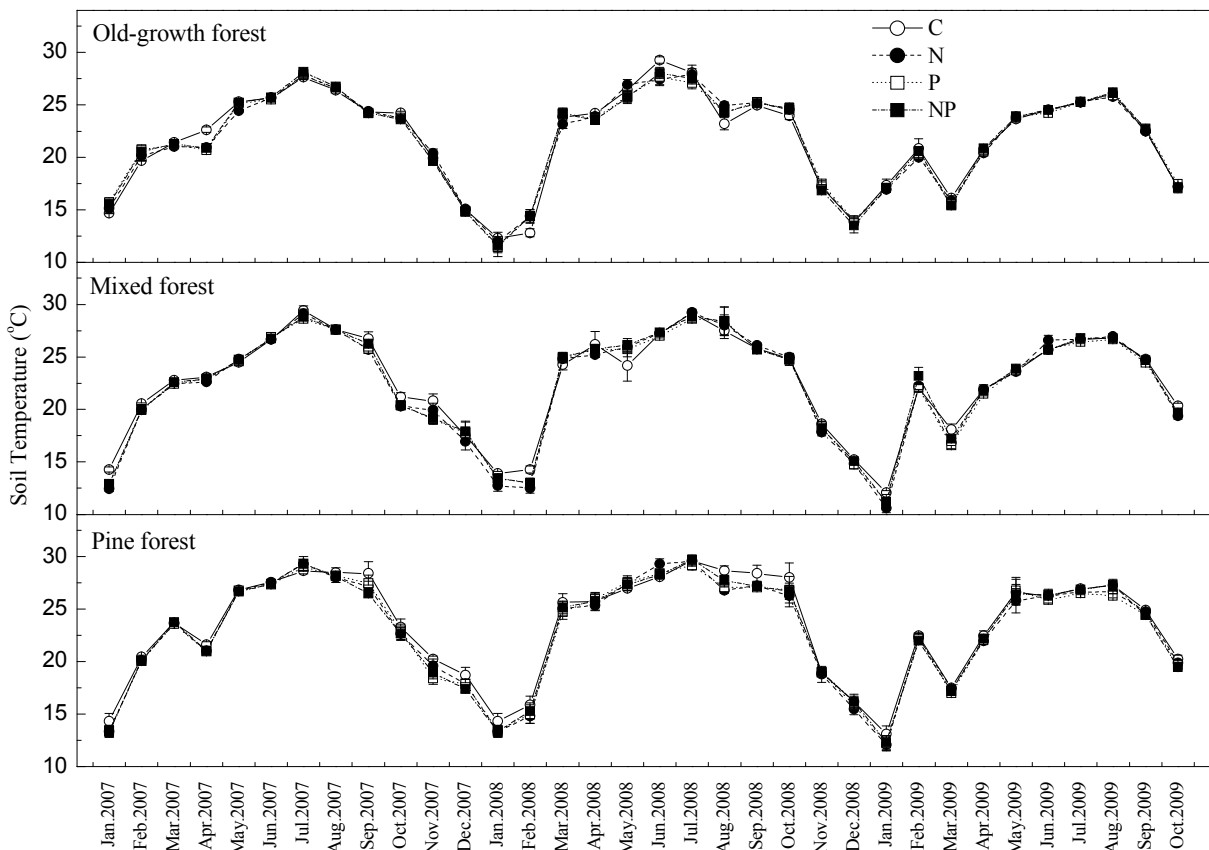

**Fig. 1** Monthly soil temperature in the three study forests of Dinghushan Biosphere Reserve (DHSBR) from January 2007 to October 2009.

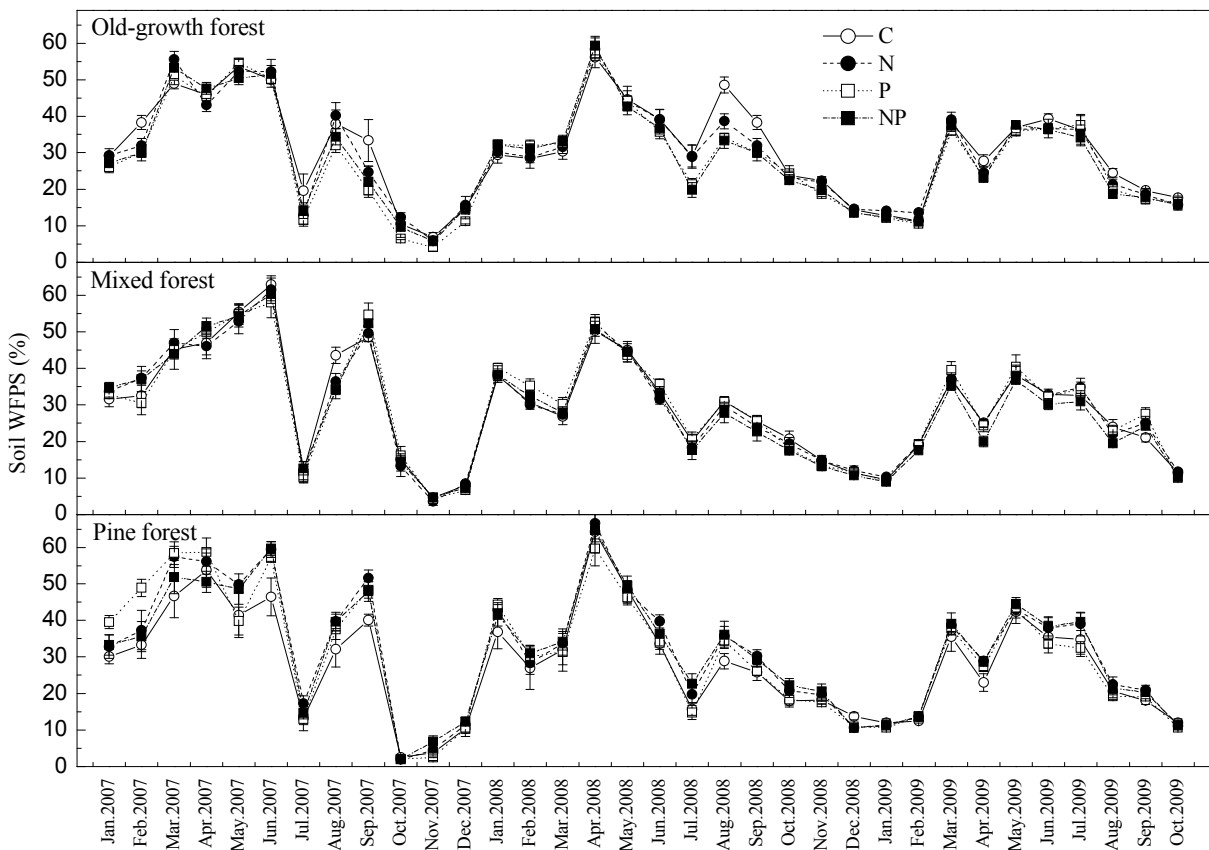

**Fig. 2** Monthly soil WFPS in the three study forests of Dinghushan Biosphere Reserve (DHSBR) from January 2007 to October 2009.

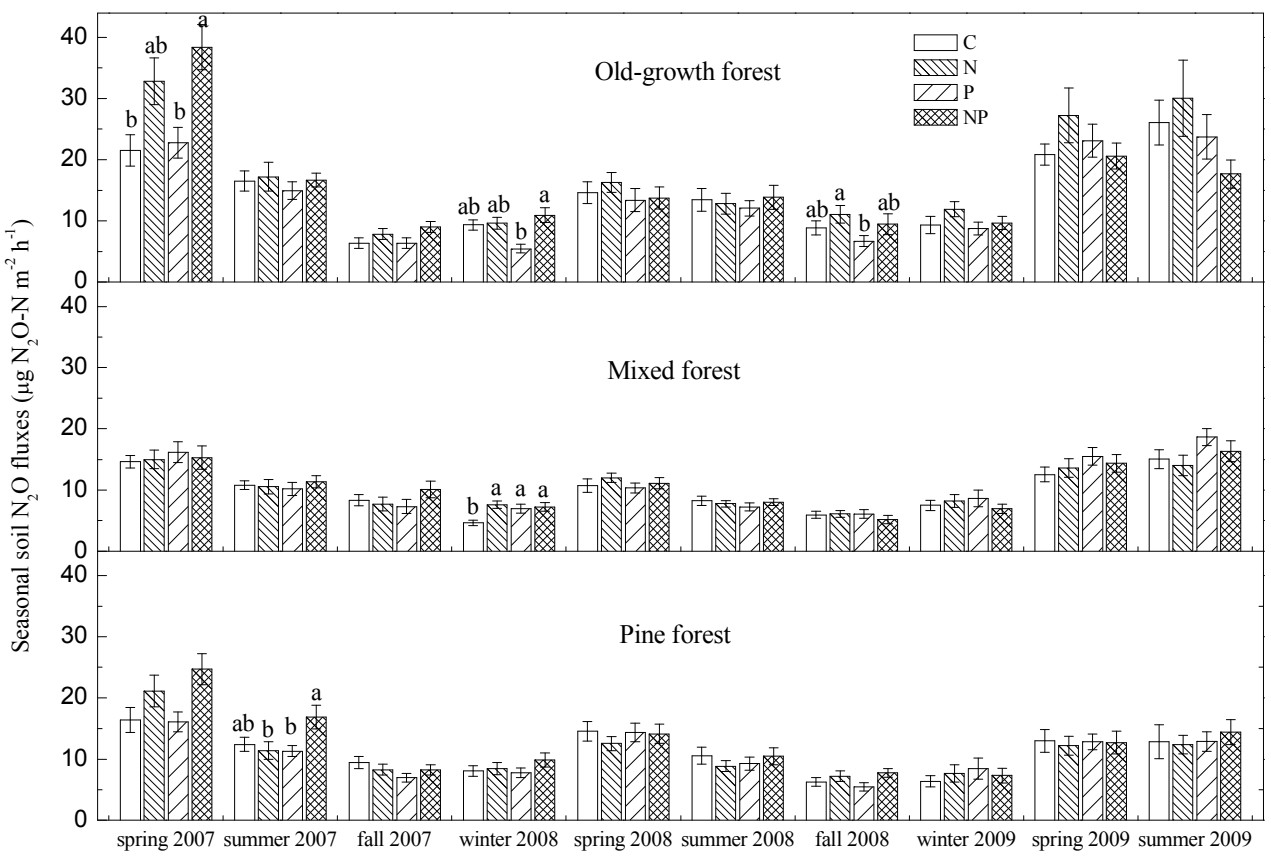

**Fig. 3** Seasonal variation of $N_2O$ fluxes in the three study forests during the sampling periods. Each error bar represents standard error of mean $N_2O$ fluxes from 5 plots (n = 5), and the data of $N_2O$ fluxes in each plot has been averaged by season (3 months). Different lowercase letters within each season represent significant differences among treatments, as determined by repeated measures ANOVA ($P < 0.05$).

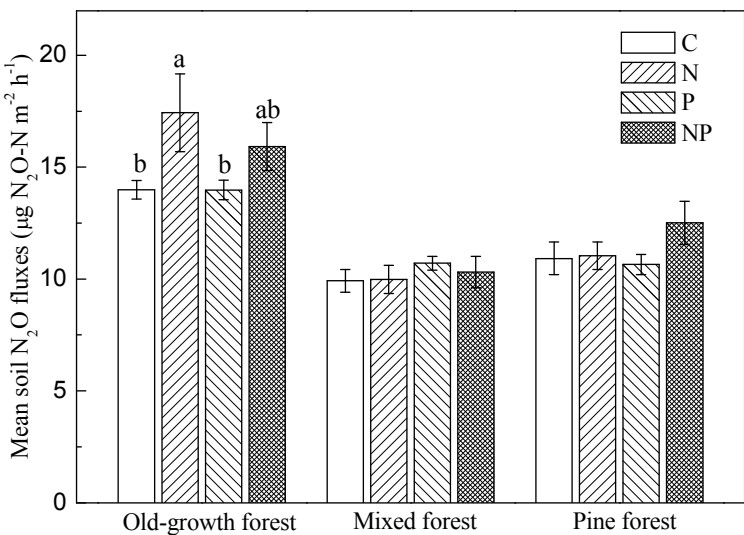

**Fig. 4** Effects of N- and P-addition on mean soil $N_2O$ fluxes from February 2007 to October 2009. Each error bar represents standard error of mean $N_2O$ fluxes from 5 plots (n = 5), and the data of $N_2O$ fluxes in each plot has been averaged from the whole sampling period (33 months). Different lowercase letters within each forest represent significant differences among treatments, as determined by repeated measures ANOVA ($P < 0.05$).

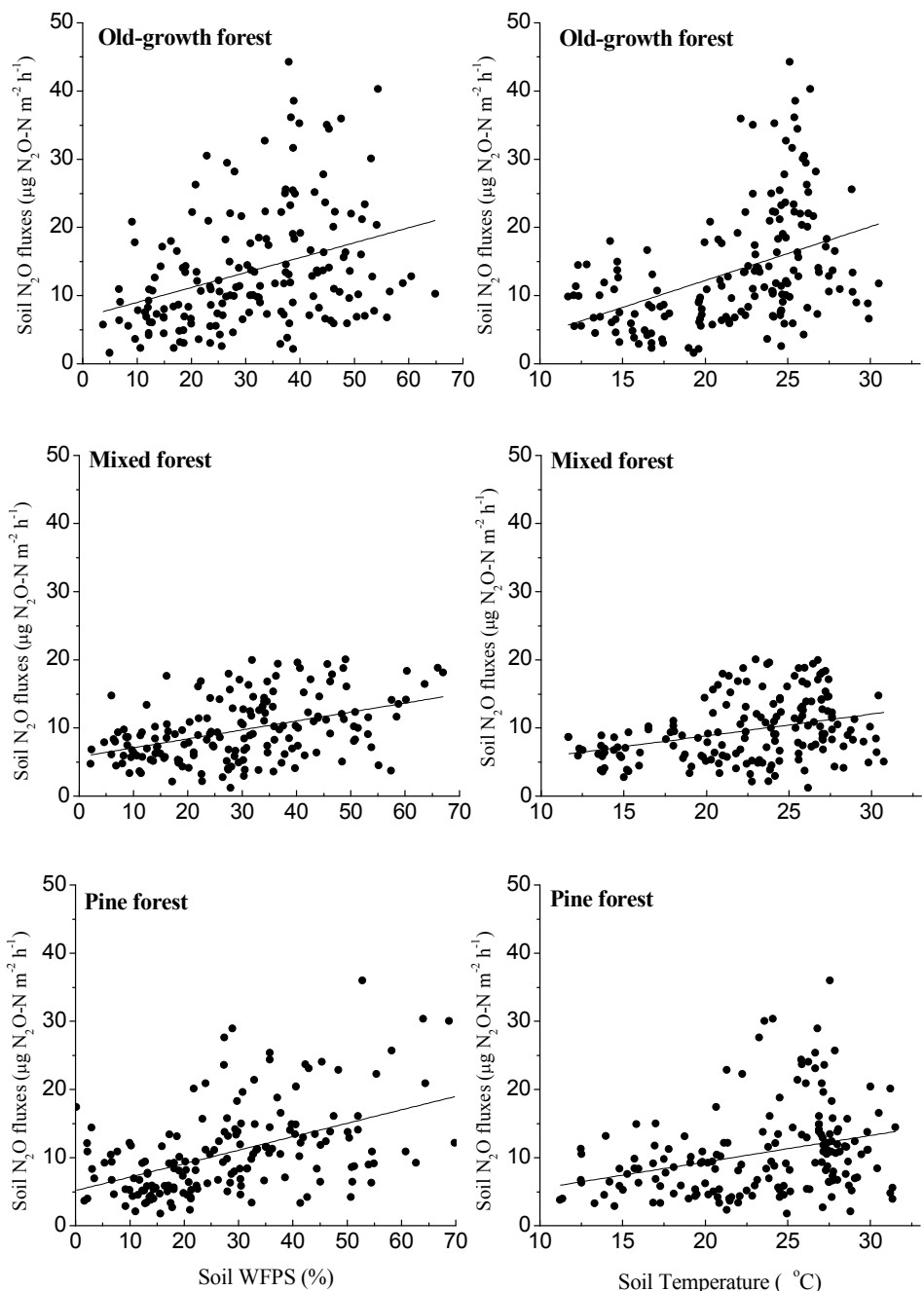

**Fig. 5** Relationships between $N_2O$ fluxes and soil WFPS (and temperature) in the five control plots of the study forests, as determined by linear regression analyses. Coefficients of the regression lines are listed in Table 4.

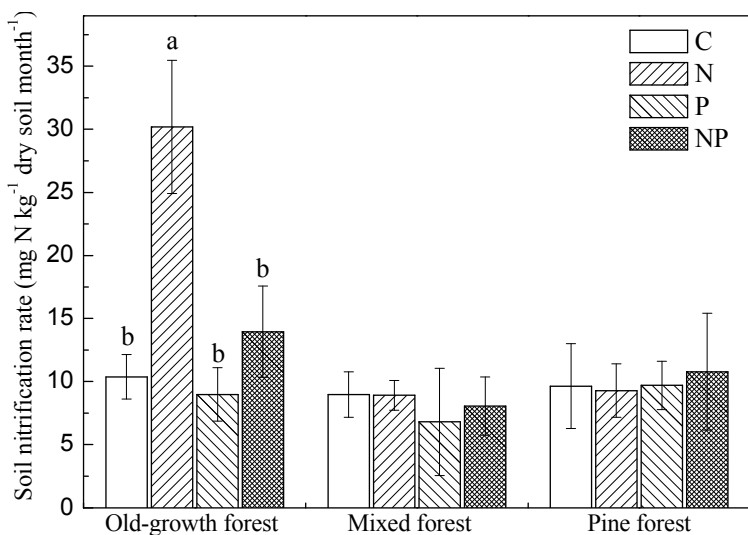

**Fig. 6** Effects of N- and P-addition on soil nitrification rate in the three study forests in August 2008. Error bars represent standard errors (n = 5). Different lowercase letters within each forest represent significant differences among treatments, as determined by one-way ANOVA ($P < 0.05$).