# Peer review of "Effects of nitrogen and phosphorus additions on nitrous oxide emission in a nitrogen-rich and two nitrogen-limited tropical forests"

_Biogeosciences, 2015_

## Referee Comment (RC1) · Anonymous Referee #2 · 19 Jan 2016

Previous comments to author earlier in the review process were considered and adequately addressed. I thank the authors for their work. I have no further comments at this time.

---

## Author Comment (AC1) · 20 Jan 2016

We would like to thank Referee #2 for the previous comments and suggestions which help us imporve the quality of the manuscript.

---

## Referee Comment (RC2) · Anonymous Referee #3 · 27 Jan 2016

This paper studied the effects of N and P additions on N2O emission in two tropical forest soils. The authors claimed that this is the first study to exam how N and P interact to control soil N2O emission in tropical forests. As far as I can tell, the results are sound, but the conclusions might need to be further discussed. I also have several technical comments, detailed below, that should be addressed prior to publication.

1) In page 6, lines 7-9, it shows that natural atmospheric N deposition is $\sim$ 50 kg N ha-1 yr-1 for this study region. Why did you add so much N (150 kg N ha-1 yr-1) for your experiments?

2) In the introduction part, it would be useful to give some information about the differences between old-growth forest and younger forest, such as soil development, plant
[Figure]

N utilization, soil N cycling, trees root. . ..

3) In page 9, lines 8-9, 'soil WFPS decreased in summer'. But in page 6, lines 4-5, you wrote 75% of precipitation falls from March to August. Why is that? Do you have the rainfall data?

4) In page 12, lines 5-6, although higher MBC in old-growth forest soil, I am still not sure about the higher activity of (de)nitrifying bacteria as the low soil pH ($\sim$ 4.0). Chemodenitrification or other chemical processes might be more important than (de)nitrification.

5) In page 14, lines 15-16, you only measured N2O emissions and nitrate leaching, but didn't measure other gases lost (NH3, NO, HONO, NO2) and also didn't measure nitrogen utilization by plant. Thus, it is hard to say that 'N continue to be utilized and was not lost . . ..', and also hard to support the hypothesis in the following sentence.

6) One way to check the mechanism of P alleviation of N2O emissions is to compare soil microbial community in Control and P addition treatments. This might give you a clue in microbiological level.

7) For my understanding, your control experiment is under natural atmospheric N deposition? Compared with control treatment, P addition treatment didn't decrease N2O flux (Fig. 3 and 4). So it is not possible to get the conclusion that 'P fertilization can be used to reduce soil N2O emission in N-rich forests under atmospheric N deposition'. Even P addition treatment decreased N2O flux compared with high N (150 kg N ha-1 yr-1) addition treatment, how do you know P addition will also decrease N2O flux under low N addition or atmospheric N deposition (50 kg N ha-1 yr-1)? Especially you explained that N2O emissions are caused by high N content or N-rich soil.

---

## Referee Comment (RC3) · Anonymous Referee #4 · 17 Feb 2016

This is a very well written paper on the impact of N and P on N2O emissions from young and old tropical forest soils. The authors carried out a statistically designed plot experiments and applied either N, P or N+P to just water to the plots and measured the N2O fluxes, soil DIN, P, SOC and microbial biomass. Their general findings, that P addition reduced the N induced N2O emissions is interesting and as the authors suggested will warrant further investigation. This paper is certainly suitable for publication in BG. There are a few mainly technical points the authors should address (see below). My only main concern is the large rates of N & P application (150 kg N / ha / y and 150 kg P / ha/ y). The N applied is ~5 times larger than the atmospheric N deposition rate at the site. The authors need to justify these unrealistic large rates. Would the results

of the paper be different if slightly more realistic rates of N and P would be have been applied? Technical comments: P7 line 1-5: you need to include a bit more detail on the chamber design: dimensions of the baseframe and lid (or chamber). Did you use a stiring fan, pressure valve? How did you seal the chamber to the lid?

P 7Line 5: Change sentence to: '. . .and analyzed within 12 h on the gas chromatograph (Agilent 4890D) fitted. . .' (replaced 'in' with 'on')

P7 line 10: 'The calculation of N2O fluxes followed the method of Holland et al. (1999), based on linear regression of' chamber gas concentration across time (changed 'across' to 'with')

P7 Line 11: was the soil temperature measured inside the chamber?

P7: line 16: I am not certain that the very general particle density value of 2.65 g/cm3 is appropriate to be used for your forest soils? Would you not expect a different particle density in the OG forest compared to the mixed/pine forests?

P8 line 3: How was NH4 extracted from the soil?

P8: line 8 & 10: NO3ǎňÂǎ- N. '-' should not be a superscript

P10 line 3:change to: ' mixed, and pine forests, respectively (Fig. 4), with being significantly higher (P = 0.001) in the old-growth forest

Page 9 line 16 delete 'were' and line 21: delete 'was'

Page 10: line 3: change to ' mixed, and pine forests, respectively (Fig. 4), with being significantly higher (P = 0.001) in the old-growth forest'

P 11 section 4.1 first paragraph: you may like to add that the variability of the data available could be due to soil type and also variability in climate

P11 line 23-24: is this the same forest as in your study? If this is the case, replace with: ...in this old-growth forest, investigated previously by Fang et al (2008)

[Figure]

P13 line 12 'In spring, forest soil was enriched with inorganic N (accumulated during non-growing seasons)'you need to say that the non growing season is due to the lack of rainfall. Also comment on the pulsing effect (wetting dry soil triggers N2O emissions and other gases.

p14 line 22: change to 'allowing us to reject the hypothesis that P addition causes greater decrease in N2O emission'

P15 line 1-2: Under laboratory conditions, Sundareshwar et al. (2003) found a negative response of sediment N2O emission to nitrate addition. This sentence should be moved to the nitrogen section 4.3

Fig 3 & 4 legend line 3: delete 'before analysis' Fig 5 legend change to: . . ."in the three control plots of the study forest. . .;

———————————————

---

## Referee Comment (RC4) · Anonymous Referee #5 · 2 Mar 2016

General comments

This study presents a field experiment studying the effects of nitrogen and phosphorus additions on soil N2O emissions from nutrient rich and nutrient poor tropical forests. In general, the paper is well written and is highly relevant, and it provides valuable new information about the combined and individual effects of N and P fertilization on soil N2O emissions. Based on three referees and the response of the authors to them, the authors have already addressed several issues related to e.g. the high N fertilization rates, effects of P on alleviation of N2O emissions, which has greatly improved the quality of the paper. However, I have few additional comments that are mainly related to the gas analysis of N2O, presentation and interpretation of the results. I consider

this work important and worth publishing after addressing the points below.

Specific comments

Page 2, lines 24-25: Could you explain how and in what respect the tropical forests have shown an increase in soil $N_2O$ emissions compared to temperate and boreal forest soils? Is this due to increased atmospheric N deposition, or do you refer merely to a pure comparison of the $N_2O$ emission rates from these ecosystems?

Page 3, lines 2-5: I consider that in addition to the mentioned factors, the poor knowledge in factors controlling $N_2O$ emissions in tropical forests is also due to the rather small number of studies from these ecosystems.

Page 3, lines 11-13: I would mention here also other losses of N, such as leaching losses, emissions of $N_2$, NO, $NH_3$, and HONO, which all are signs of N saturation.

Page 5, lines 18-20: I would like to see here more description of where the N is retained (soil, above ground biomass, below ground biomass, microbial biomass), and in what forms are the N losses from the soil (leaching, gaseous losses, what gas species etc).

Page 7, lines 5-7: could you give more details of the gas chromatographic analysis. I'm missing information of the used columns, oven temperature, flow rates, carrier and make-up gases. Especially, I'm interested and slightly concerned whether $CO_2$ was allowed to enter the ECD, or whether it was trapped chemically (e.g. ascarite) as if $N_2$ is used as a carrier gas, and $CO_2$ is allowed to enter the ECD, this may bias the $N_2O$ analysis and lead to overestimated $N_2O$ fluxes as described by Zheng et al. (2008).

Zheng X., et al., 2008. Quantification of $N_2O$ fluxes from soil-plant systems may be biased by the applied gas chromatograph methodology. Plant and Soil, 311: 211-234.

Page 8, lines 13-19: I'm missing information whether you tested the data for normality and equality of variances. Naturally, if these criteria were met, the use of parametric tests are justified, otherwise non-parametric tests should be used. Please, clarify this.

Page 12, lines 13-14: This line is almost identical to the sentence from page 8, lines 23-24. Please, modify.

Page 12, lines 16-18: was the difference in mean soil temperature statistically significant between the three forests? If yes, please give the p-value. Also, were the N2O emission rates across different forests significantly different? If yes, please give the p-value here. In other words, if the above mentioned differences were not statistically significant, you cannot claim that soil temperature does not explain the N2O emission pattern across the forests.

Page 12-13, chapter 4.2: You present simple correlation analysis of N2O emissions against soil temperature or soil moisture, and use robust linear regression to explain the N2O emissions (Fig. 5). Based on the scatter plots, it seems that there is an exponential relationship between at least N2O fluxes and soil temperature. Did you try to fit also non-linear models to the data? Also, as the correlation between both N2O flux and soil temperature, and N2O flux and soil moisture are highly significant, did you try to build a regression model including both soil temperature and soil moisture as parameters? This might be worth the effort.

Page 13, lines 1-7: Based on only two soil sampling occasions (Feb 2007 and Aug 2009) it is very uncertain to conclude how the soil inorganic N concentrations developed during the different seasons. For instance, a soil sampling in February 2007 does not support that the soil was enriched with inorganic N, and also a soil sampling in August 2009 does not support that the inorganic N had decreased during the growing season, as there were no measurements during the growing season. Please, discuss these uncertainties, and if possible bring in material and references to support your conclusions.

Table 2 and e.g. page 16, lines 18-20: The values in soil pH, inorganic N, organic C, microbial biomass and P in Table 2 are only from one sampling occasion, approximately two years from the start of the experiment. Also, the comparison between the

fertilization treatments is conducted with data from one time sampling only, while the fertilization was conducted every second week over a two-year period. I see here a problem when comparing the effects of the fertilization. Firstly, I think it would be best to compare the soil N (and other measures) status before and after the treatments. But in this comparison, the timing of the sampling is important as the soil N (and other) have strong seasonality, which may be larger than the treatment effect. As the soil sampling before the experiment was in the spring (February 2007), and the soil sampling after the experiment was during summer (August 2009), it is very difficult to know whether the differences result from the treatments or the seasonal variation in soil N. My other concern is that the different plots may have differed between each other already before the experiment. Did you test this? Overall, I think it is very difficult to conclude that the fertilization did or did not influence the soil N status in the experiment. Please, discuss these uncertainties or be more careful in interpreting the results, unless there is more data to support these findings.

Fig. 5: Is this data from the control plots only? Please, specify which data was used.

Technical corrections

Page 2, line 12: add "atmospheric lifetime" inside the parenthesis. Page 4, line 15, and line 19: change a N-rich to "an N-rich" Page 6, line 7: I assume that you mean wet N deposition. If so, please add the word "wet" to the "Inorganic N deposition. . .". Or if this is a sum of wet and dry deposition, please clarify it. Page 8, line 25; page 9, line 9; page 10, line 2 and elsewhere in the paper: I would harmonize the use of decimal places, preferably round them to one decimal place. At least with N2O fluxes, I don't think the precision of the measurement is high enough to give the emissions with the accuracy of two decimal places. Page 12, line 22: add "WFPS" and "the" to the sentence: "highest WFPS in the old-growth forest and the lowest in the pine forest" Tables 1 and 2. Please, give the numbers with one decimal place.

---

## Author Response (AR1)

Dear Associate Editor,

We would like to thank four anonymous referees for their constructive comments and suggestions during the Interactive Discussion, and we also thank you for your decision of giving us the opportunity to revise this manuscript. All the comments and suggestions help us greatly improve the quality of the manuscript. My co-authors and I agree with the comments, and we have improved the manuscript according to the comments during the Interactive Discussion. We have also checked the response letter and the revised manuscript again after the Interactive Discussion was closed. Below are our detailed point-by-point responses. All the revised portions are marked in red in the revised manuscript, and the page and line numbers of the revised manuscript are also provided. If you have any more questions, please do not hesitate to contact me. Thank you very much.

Best regards,
Jiangming Mo
Email: mojm@scib.ac.cn

**Referee #2**

Previous comments to author earlier in the review process were considered and adequately addressed. I thank the authors for their work. I have no further comments at this time.
**Answer:** We would like to thank Referee #2 for the previous comments and suggestions on the manuscript.

**Referee #3**

This paper studied the effects of N and P additions on $N_2O$ emission in two tropical forest soils. The authors claimed that this is the first study to exam how N and P interact to control soil $N_2O$ emission in tropical forests. As far as I can tell, the results are sound, but the conclusions might need to be further discussed. I also have several technical comments, detailed below, that should be addressed prior to publication.
**Answer:** Thank you for these comments.

1) In page 6, lines 7-9, it shows that natural atmospheric N deposition is ~50 kg N ha$^{-1}$ yr$^{-1}$ for this study region. Why did you add so much N (150 kg N ha$^{-1}$ yr$^{-1}$) for your experiments?
**Answer:** Thank you for pointing out this question, and Referee #4 also pointed out the similar question that why we used the high rates of N and P fertilization.

In fact, we have another N addition experiment in the old-growth forest using different N gradients (50, 100, and 150 kg N ha$^{-1}$ yr$^{-1}$) which are 1−3 folds of atmospheric N deposition rate (~50 kg N ha$^{-1}$ yr$^{-1}$), and we found that many soil processes responded significantly only following high N addition (150 kg N ha$^{-1}$ yr$^{-1}$) in this forest. For example, our previous studies found that only high N addition significantly decreased soil respiration rates (Mo et al., 2008), methane uptake rates (Zhang et al., 2008), fine root biomass and soil pH (Lu et al., 2010) in the old-growth forest. These results suggest that soil processes may have a high N threshold in this N-rich forest. Although the two younger forests are N-limited, we used a similar N gradient (150 kg N ha$^{-1}$ yr$^{-1}$) for the main purpose of comparison among the three forests (Zheng et al., 2015; Zhu et al., 2013). Secondly, we used the high P addition rate because of the high P demand of soil microbes in our old-growth forest (Liu et al., 2012). The high fertilization rates (150 kg N ha$^{-1}$ yr$^{-1}$ and 150 kg P ha$^{-1}$ yr$^{-1}$) can remove all possible N and P constraints in both young and old-growth forests (Cleveland and Townsend, 2006). Finally, our experiment design (including the plot size and fertilizer level) also refers to the experiment in a tropical forest in Costa Rica (Cleveland and Townsend, 2006).

Thus, to clearly clarify why we used the high fertilization rates, we have added these information in the Materials and Method section: "We used the high N gradient, about 3 folds of atmospheric N deposition rate, because many soil processes responded significantly only under this gradient in the old-growth forest (Mo et al., 2008; Zhang et al., 2008a; Lu et al., 2010). High P gradient was used because of the high P demand of soil microbes in the old-growth forest (Liu et al., 2012). Although the two younger forests are N-limited, we used the similar N and P gradients for the main purpose of comparison among the forests (Zheng et al., 2015; Zhu et al., 2013a). High fertilization rates can remove all possible N and P constraints in both young and old-growth forests (Cleveland and Townsend, 2006). In addition, plot size and fertilizer level in our forests were also the same as those in Costa Rica by Cleveland and Townsend (2006)." (Please also see Page 7 Lines 8-15 in the revised manuscript).

Reference:
Cleveland, C. C., and Townsend, A. R.: Nutrient additions to a tropical rain forest drive substantial soil carbon dioxide losses to the atmosphere, P. Natl. Acad. Sci. USA, 103, 10316-10321, 2006.
Liu, L., Gundersen, P., Zhang, T., and Mo, J. M.: Effects of phosphorus addition on soil microbial biomass and community composition in three forest types in tropical China, Soil Biol. Biochem., 44, 31-38, 2012.
Mo, J., Zhang, W., Zhu, W., Gundersen, P., Fang, Y., Li, D., and Wang, H.: Nitrogen addition reduces soil respiration in a mature tropical forest in southern China, Global Change Biol., 14, 403-412, 2008.
Zhang, W., Mo, J., Zhou, G., Gundersen, P., Fang, Y., Lu, X., Zhang, T., and Dong, S.: Methane uptake responses to nitrogen deposition in three tropical forests in southern China, J. Geophys. Res, 113, 2008.
Zhu, F. F., Yoh, M., Gilliam, F. S., Lu, X. K., and Mo, J. M.: Nutrient limitation in

three lowland tropical forests in southern China receiving high nitrogen deposition: insights from fine root responses to nutrient additions, PLoS One, 8, e82661, 2013.

Zheng, M., Huang, J., Chen, H., Wang, H., and Mo, J.: Responses of soil acid phosphatase and beta-glucosidase to nitrogen and phosphorus addition in two subtropical forests in southern China, Eur. J. Soil Biol., 68, 77-84, 2015.

2) In the introduction part, it would be useful to give some information about the differences between old-growth forest and younger forest, such as soil development, plant N utilization, soil N cycling, trees root….

**Answer:** Thank you for this good suggestion, and the information you provided will help our manuscript clarify why studied on different forest types (old-growth forest versus younger forest) more clearly.

Accordingly, we have followed your suggestion and added this information in the Introduction section: "However, the capacity of P to reduce N losses may be related to forest development. Despite many tropical forests have rich N in soils, several younger forests early in soil development are still N-limited (Vitousek et al., 1997a). Compared with the old-growth forests, younger forests often show the higher N demands and utilization of plants and microbes, but the lower rates of soil N cycling, such as mineralization, nitrification and leaching (Aber et al., 1998). In contrast, old-growth forests have the higher P demand because they are commonly depleted in P (Vitousek et al., 2010). For example, one of our previous study showed that soil microbes and/or tree roots released more phosphatase in the old-growth forest than in the younger one (Zheng et al., 2015)." (Please also see Page 4 Lines 22-25 and Page 5 Lines 1-4 in the revised manuscript).

Reference:

Aber, J., McDowell, W., Nadelhoffer, K., Magill, A., Berntson, G., Kamakea, M., McNulty, S., Currie, W., Rustad, L., and Fernandez, I.: Nitrogen saturation in temperate forest ecosystems-Hypotheses revisited, Bioscience, 48, 921-934, 1998.

Vitousek, P. M., and Farrington, H.: Nutrient limitation and soil development: experimental test of a biogeochemical theory, Biogeochemistry, 37, 63-75, 1997a.

Vitousek, P. M., Porder, S., Houlton, B. Z., and Chadwick, O. A.: Terrestrial phosphorus limitation: mechanisms, implications, and nitrogen-phosphorus interactions, Ecol. Appl., 20, 5-15, 2010.

Zheng, M., Huang, J., Chen, H., Wang, H., and Mo, J.: Responses of soil acid phosphatase and beta-glucosidase to nitrogen and phosphorus addition in two subtropical forests in southern China, Eur. J. Soil Biol., 68, 77-84, 2015.

3) In page 9, lines 8-9, 'soil WFPS decreased in summer'. But in page 6, lines 4-5, you wrote 75% of precipitation falls from March to August. Why is that? Do you have the rainfall data?

**Answer:** Thank you for pointing out this interesting question, and we have made some improvements in the text.

First, rainfall (precipitation) is not the main data supporting the mechanisms in our study, so we did not measure it. However, previous study showed that 75% of precipitation fell in spring (March to May) and summer (June to August) in the study forests (Huang and Fan, 1982), and our recent study also indicated a similar pattern (73.5%) (Lu et al., 2013), suggesting that precipitation pattern changes little in spite of slight fluctuation in the study region. Thus, to make this more clearly, we have replaced "75% of which falls from March to August…." with "about 75% of which falls from March to August…. as reported by our previous studies (Huang and Fan, 1982; Lu et al., 2013)" in the text (Please also see Page 6 Lines 18-20 in the revised manuscript).

Second, why soil WFPS was high in spring but decreased in summer? We infer this may be caused by the higher plant uptake and transpiration in the summer. Plants may grow fast in the summer (growing season), and thus absorb more water directly from soils or the soil nutrients which are also carried by water. A recent study carried out by our colleagues found that dominant tree species in this forest generally showed the higher sap flow velocity and daily transpiration in the summer than in the dry season (Cheng et al., 2015). Thus, we have added this information in the Results section: "possibly due to the higher plant uptake and transpiration, despite the high amount of precipitation in summer" (Please also see Page 10 Line 23 in the revised manuscript), and also in the Discussion section: "In summer, $N_2O$ emission began to decrease given decreasing soil WPFS (Fig. 3) possibly caused by the higher plant uptake and transpiration (Cheng et al., 2015)." (Please also see Page 15 Lines 2-3 in the revised manuscript).

Reference:

Cheng J., Oyang X., Huang D. W., Liu S. Z., Zhang D. Q., Li Y. L.: Sap flow characteristics of four dominant tree species in a mixed conifer broadleaf forest in Dinghushan, Acta Ecologica Sinica, 35, 4097-4104, 2015.

Huang, Z. F., and Fan, Z. G.: The climate of Dinghushan (in Chinese with English abstract), Tropical and Subtropical Forest Ecosystem, 1, 11-16, 1982.

Lu, X., Gilliam, F. S., Yu, G., Li, L., Mao, Q., Chen, H., and Mo, J.: Long-term nitrogen addition decreases carbon leaching in nitrogen-rich forest ecosystems, Biogeosciences, 10, 3931-3941, 2013.

4) In page 12, lines 5-6, although higher MBC in old-growth forest soil, I am still not sure about the higher activity of (de)nitrifying bacteria as the low soil pH (~4.0). Chemodenitrification or other chemical processes might be more important than (de)nitrification.

**Answer:** Thank you. We agreed with your comments. We have looked up some relevant references, and understood that biological (de)nitrification are often active under neutral and slightly alkaline conditions, while chemo-denitrification is more important than biological (de)nitrification in acid conditions, especially when soil pH

was lower than 4.0 (Tate, 1995; Chalk and Smith, 1983; Mørkved et al., 2007).

Accordingly, the activity of (de)nitrifying bacteria may not be a proper explanation for the higher $N_2O$ emission in our old-growth forest soil which is acid, so we have replaced "Compared to the two younger forests, the old-growth forest had significantly higher soil dissolved organic C, total organic C, and microbial biomass C (Table 1), likely supporting a higher activity of nitrifying and denitrifying bacteria responsible for $N_2O$ production" with "Additionally, the old-growth forest had significantly higher soil dissolved organic C and total organic C (Table 1), which could provide more C energy for $N_2O$ production (Zhang et al., 2014)." in the text (Please also see Page 13 Lines 24-25 in the revised manuscript).

In addition, following your suggestion, we have also added another explanation: "Compared to the younger forests, the old-growth forest had more acid soil conditions (Table 1 and 2), likely supporting the higher chemo-denitrification (Tate, 1995; Chalk and Smith, 1983; Mørkved et al., 2007)." (Please also see Page 13 Lines 22-24 in the revised manuscript).

Reference:

Chalk, P., and Smith, C.: Chemodenitrification, in: Gaseous loss of nitrogen from plant-soil systems, Springer, 65-89, 1983.

Mørkved, P. T., Dörsch, P., and Bakken, L. R.: The $N_2O$ product ratio of nitrification and its dependence on long-term changes in soil pH, Soil Biol. Biochem., 39, 2048-2057, 2007.

Tate, R. L.: Soil microbiology, John Wiley and Sons, 398, 1995.

Zhang, W., Zhu, X., Luo, Y., Rafique, R., Chen, H., Huang, J., and Mo, J.: Responses of nitrous oxide emissions to nitrogen and phosphorus additions in two tropical plantations with N-fixing vs. non-N-fixing tree species, Biogeosciences, 11, 4941-4951, 2014.

5) In page 14, lines 15-16, you only measured $N_2O$ emissions and nitrate leaching, but didn't measure other gases lost ($NH_3$, NO, HONO, $NO_2$) and also didn't measure nitrogen utilization by plant. Thus, it is hard to say that 'N continues to be utilized and was not lost….', and also hard to support the hypothesis in the following sentence.

**Answer:** Thank you very much for pointing out this question, and we agreed with this constructive comment. Although we did not measure other gases lost ($NH_3$, NO, HONO, $NO_2$) and the nitrogen utilization by plant, our previous studies have showed that the N input was mainly lost via leaching or retained for plant biomass and litter increment in the two younger forests, as explained below.

First, our previous survey showed that atmospheric N deposition via precipitation was 49.5 kg N $ha^{-1}$ $yr^{-1}$ for this region, and total dissolve N leaching losses (surface runoff plus seepage leaching in soil solution) from the upper 20 cm soil was 29 and 22 kg N $ha^{-1}$ $yr^{-1}$ for the pine and mixed forest, respectively, and 21 and 28 kg N $ha^{-1}$ $yr^{-1}$ (for the pine and mixed forest, respectively) was retained in the upper 20cm soil and through plant uptake (Fang et al., 2008). These retention estimates based on input-output budgets also account for the potential gaseous N loss by (de)nitrification

(Fang et al., 2008).

Second, in the pine forest, previous estimate suggests that the canopy tree, the understory plants and standing floor litter accumulated 9.1, 6.0 and 6.5 kg N ha$^{-1}$ yr$^{-1}$, respectively, during the period from 1990 to 2000 (Mo et al., 2004), and these total N accumulation approximates to the observed 21 kg N ha$^{-1}$ yr$^{-1}$ that was retained above the upper 20 cm soil (Fang et al., 2008). In the mixed forest, N accumulation in the plant biomass and the increasing litter layer were probably higher than in the pine forest, due to higher litter production and higher foliar N concentration (Mo et al., 2007), and might as well account for the 28 kg N ha$^{-1}$ yr$^{-1}$ retained in this forest (Fang et al., 2008). Thus, under N deposition, the N retention in two younger forests was in accordance with the estimates of N accumulation in biomass and litter increment (Fang et al., 2008), suggesting that the N input had less effect on gaseous N loss in the two younger forests.

Third, our N addition study showed that N addition had no effects on nitrification rate and N$_2$O emission in the younger forests (Fig. 4 and 6), further suggesting that the N retention was mainly used for plant growth rather than for gaseous N loss.

The above evidences may support "N continues to be utilized following N addition", but we admit that "N was not lost" is not true because parts of the added N were lost via leaching (Fang et al., 2008). Thus, we have made some improvements in the text: (1) we have added this information: "In addition, our previous study showed that under atmospheric N deposition, the N retention in the two forests was in accordance with the estimates of N accumulation in plant biomass and litter increment (Mo et al., 2004, 2007a; Fang et al., 2008), suggesting that the N retention was mainly used for plant growth rather than gaseous N loss." (Please also see Page 16 Lines 11-14 in the revised manuscript); (2) we have added this statement: "In this study, despite we did not measure other gases losses (such as NH$_3$, NO, HONO and NO$_2$) which are also important in forest soils,…." (Please also see Page 16 Lines 14-15 in the revised manuscript).; (3) we have replaced this statement "N continues to be utilized and was not lost" with "N continues to be utilized rather than N$_2$O emission" (Please also see Page 16 Lines 17-18 in the revised manuscript).; (4) we have deleted this hypothesis in the text: "This confirms our hypothesis that soil N$_2$O emission shows no response to N addition in N-limited forests (Zhang et al., 2008)."; and (5) we have added this information: "Further studies are needed to examine whether N addition increases other nitrogenous gases loss in the N-limited forests." (Please also see Page 16 Lines 18-19 in the revised manuscript).

Reference:

Fang, Y. T., Gundersen, P., Mo, J. M., and Zhu, W. X.: Input and output of dissolved organic and inorganic nitrogen in subtropical forests of South China under high air pollution, Biogeosciences, 5, 339-352, 2008.

Mo, J. M., Peng, S. L., Brown, S., Kong, G. H., and Fang, Y. T.: Nutrient dynamics in response to harvesting practices in a pine forest of subtropical China, Acta Phytoecol. Sin., 28, 810–822, 2004.

Mo, J. M., Brown, S., Xue, J. H., Fang, Y. T., Li, Z. A., Li, D. J., and Dong, S. F.:

Response of nutrient dynamics of decomposing pine (Pinus massoniana) needles to simulated N deposition in a disturbed and a rehabilitated forest in tropical China, Ecol. Res., 22, 649–658, 2007a

6) One way to check the mechanism of P alleviation of $N_2O$ emissions is to compare soil microbial community in Control and P addition treatments. This might give you a clue in microbiological level.

**Answer:** Thank you for this good suggestion, and we have added the relevant information on soil microbial community to support the mechanism of P alleviation of $N_2O$ emissions. We have added: "P addition likely alleviated the P limitation on soil microbes in our old-growth forest, because our previous study showed that P addition significantly increased soil microbial biomass and soil respiration (Liu et al., 2012). Compared with the controls, P addition changed soil microbial community, including the increases in biomass of bacteria and AM fungi (Liu et al., 2012, 2013). The increases in AM fungi may help plants acquire more N and P nutrients (Tresede and Vitousek, 2001), because they are more efficient in obtaining nutrients from the soil than the plant roots (Liu et al., 2013). In addition, the increases in soil bacterial and fungal biomass may potentially increase total N acquirement, as evidenced by our previous study showing that 4 years of P- and NP-addition tended to increase soil microbial biomass N (Liu et al., 2013)." (Please also see Page 18 Lines 17-24 in the revised manuscript).

Reference:

Liu, L., Gundersen, P., Zhang, T., and Mo, J. M.: Effects of phosphorus addition on soil microbial biomass and community composition in three forest types in tropical China, Soil Biol. Biochem., 44, 31-38, 2012.

Liu, L., Zhang, T., Gilliam, F. S., Gundersen, P., Zhang, W., Chen, H., and Mo, J. M.: Interactive effects of nitrogen and phosphorus on soil microbial communities in a tropical forest, PLoS One, 8, e61188, 2013.

Treseder, K. K., and Vitousek, P. M.: Effects of soil nutrient availability on investment in acquisition of N and P in Hawaiian rain forests, Ecology, 82, 946-954, 2001.

7) For my understanding, your control experiment is under natural atmospheric N deposition? Compared with control treatment, P addition treatment didn't decrease $N_2O$ flux (Fig. 3 and 4). So it is not possible to get the conclusion that 'P fertilization can be used to reduce soil $N_2O$ emission in N-rich forests under atmospheric N deposition'. Even P addition treatment decreased $N_2O$ flux compared with high N (150 kg N ha$^{-1}$ yr$^{-1}$) addition treatment, how do you know P addition will also decrease $N_2O$ flux under low N addition or atmospheric N deposition (50 kg N ha$^{-1}$ yr$^{-1}$)? Especially you explained that $N_2O$ emissions are caused by high N content or N-rich soil.

**Answer:** Thank you for pointing out this excellent question, and we agreed with your comments. It is interesting that P addition treatment decreased $N_2O$ emission compared with high N addition treatment (150 kg N ha$^{-1}$ yr$^{-1}$), but not when compared

with natural atmospheric N deposition (50 kg N ha$^{-1}$ yr$^{-1}$). We suggested two following reasons accounting for this phenomenon.

First, this may be related to the levels of N addition. It is possible that low N addition (or natural atmospheric N deposition) may not cause a significant increase in N$_2$O emission in this N-rich forest soil. Our previous study showed that N loss via leaching in our N-rich forest was higher than the N input via atmospheric deposition (~50 kg N ha$^{-1}$ yr$^{-1}$), suggesting a net N loss under this low N input conditions and thus the less N retained for N$_2$O production (Fang et al., 2008). Accordingly, it is possible that low N addition fail to increase soil N$_2$O emission in this N-rich forest, and P addition may show no alleviated effect. In contrast, under high N addition (150 kg N ha$^{-1}$ yr$^{-1}$), apart from N leaching, parts of the N input may be used for increasing N$_2$O emission, and thus P addition may show the alleviated effect.

Second, a lack of response of N$_2$O emission to P addition compared with the control may also be related to fertilization period. Because nutrients (N and P) addition in our study was only applied for about 2 years, we did not observe the alleviated effect of P addition on N$_2$O emission under natural N deposition (Fig. 3 and 4). However, our recent study in the same forest found that long-term (6 years) P addition (150 kg P ha$^{-1}$ yr$^{-1}$) significantly decreased soil N$_2$O emission compared with the control (natural N deposition) (Chen et al., 2015). This in part suggests that fertilization period is also an important factor affecting the alleviated effect of P addition on N$_2$O emission in this N-rich forest.

Thus, based on your suggestion and the above two possible reasons, we have made some improvements in the text: (1) we have added this information: "It is interesting that soil N$_2$O emission reduced after P addition compared with that after N addition (150 kg N ha$^{-1}$ yr$^{-1}$), but not when compared with that under atmospheric N deposition (~50 kg N ha$^{-1}$ yr$^{-1}$). We infer this may be related to the levels of N addition and/or the period of P addition. First, it is possible that low N addition, such as atmospheric N deposition in our study, may not cause a significant increase in soil N$_2$O emission in this N-rich forest. Our previous study showed that under atmospheric N deposition (49.5 kg N ha$^{-1}$ yr$^{-1}$), soil had higher N leaching (59.8 kg N ha$^{-1}$ yr$^{-1}$) in this N-rich forest, suggesting a net N loss under atmospheric N deposition (low N input), and thus the less N retained for N$_2$O production (Fang et al., 2008). Accordingly, it is possible that low N addition fail to increase soil N$_2$O emission in the N-rich forest, and thus P addition may show no alleviated effect. Second, a lack of response of N$_2$O emission to P addition compared with the control may also be related to the P fertilization period. Nutrients (N and P) addition in our study was only applied for about 2 years, and we did not observe the alleviated effect of P addition on soil N$_2$O emission under atmospheric N deposition (Fig. 3 and 4). However, our recent study in the same forest found that long-term (6 years) P addition significantly decreased soil N$_2$O emission compared with the control (atmospheric N deposition) (Chen et al., 2015). This suggests that fertilization period is also an important factor affecting the alleviated effect of P addition on N$_2$O emission in this N-rich forest." (Please also see Page 19 Lines 4-18 in the revised manuscript); (2) we have replaced this statement "Therefore, our findings suggest that P addition will alleviate the

stimulating effects of N on N$_2$O emission in the N-rich forest." with "Therefore, our findings suggest that P addition will alleviate the stimulating effects of N on N$_2$O emission in the N-rich forest, but this effect may only occur under high N addition and/or long-term P addition." (Please also see Page 19 Lines 19-20 in the revised manuscript); and (3) we have also added this sentence "this effect may only occur under high N deposition and/or long-term P addition," in the Abstract and Conclusions section (Please also see Page 2 Lines 8-9 and Page 20 Lines 3-4 in the revised manuscript).

Reference:

Fang, Y. T., Gundersen, P., Mo, J. M., and Zhu, W. X.: Input and output of dissolved organic and inorganic nitrogen in subtropical forests of South China under high air pollution, Biogeosciences, 5, 339-352, 2008.

Chen, H., Gurmesa, G. A., Zhang, W., Zhu, X., Zheng, M., Mao, Q., Zhang, T., and Mo, J.: Nitrogen saturation in humid tropical forests after 6 years of nitrogen and phosphorus addition: Hypothesis testing, Funct. Ecol., doi: 10.1111/1365-2435.12475, 2015.

**Referee #4**

This is a very well written paper on the impact of N and P on N$_2$O emissions from young and old tropical forest soils. The authors carried out a statistically designed plot experiments and applied either N, P or N+P to just water to the plots and measured the N$_2$O fluxes, soil DIN, P, SOC and microbial biomass. Their general findings, that P addition reduced the N induced N$_2$O emissions is interesting and as the authors suggested will warrant further investigation. This paper is certainly suitable for publication in BG. There are a few mainly technical points the authors should address (see below).

**Answer:** Thank you very much for these positive comments.

1. My only main concern is the large rates of N & P application (150 kg N / ha / y and 150 kg P / ha/ y). The N applied is ~5 times larger than the atmospheric N deposition rate at the site. The authors need to justify these unrealistic large rates. Would the results of the paper be different if slightly more realistic rates of N and P would be have been applied?

**Answer:** Thank you for pointing out this question. In our study region, inorganic N deposition is about 24−34 kg N ha$^{-1}$yr$^{-1}$, with an additional input of 15−20 kg N ha$^{-1}$yr$^{-1}$ as dissolved organic N, so atmospheric N deposition is about 50 kg N ha$^{-1}$yr$^{-1}$ (Page 6 Line 22−23 in the revised manuscript).

First, we have another N addition experiment in the old-growth forest using different N gradients (50, 100, and 150 kg N ha$^{-1}$ yr$^{-1}$) which are 1−3 folds of atmospheric N deposition rate (~50 kg N ha$^{-1}$ yr$^{-1}$), and we found that many soil

processes responded significantly only following high N addition (150 kg N ha$^{-1}$ yr$^{-1}$) in this forest. For example, our previous studies found that only high N addition significantly decreased soil respiration rates (Mo et al., 2008), methane uptake rates (Zhang et al., 2008), fine root biomass and soil pH (Lu et al., 2010) in the old-growth forest. These results suggest that soil processes may have a high N threshold in this N-rich forest. Although the two younger forests are N-limited, we used a similar N gradient (150 kg N ha$^{-1}$ yr$^{-1}$) for the main purpose of comparison among the three forests (Zheng et al., 2015; Zhu et al., 2013). Secondly, we used the high P addition rate because of the high P demand of soil microbes in our old-growth forest (Liu et al., 2012). The high fertilization rates (150 kg N ha$^{-1}$ yr$^{-1}$ and 150 kg P ha$^{-1}$ yr$^{-1}$) can remove all possible N and P constraints in both young and old-growth forests (Cleveland and Townsend, 2006). Finally, our experiment design (including the plot size and fertilizer level) also refers to the experiment in a tropical forest in Costa Rica (Cleveland and Townsend, 2006).

Thus, to clearly clarify why we used the high fertilization rates, we have added these information in the Materials and Method section: "We used the high N gradient, about 3 folds of atmospheric N deposition rate, because many soil processes responded significantly only under this gradient in the old-growth forest (Mo et al., 2008; Zhang et al., 2008a; Lu et al., 2010). High P gradient was used because of the high P demand of soil microbes in the old-growth forest (Liu et al., 2012). Although the two younger forests are N-limited, we used the similar N and P gradients for the main purpose of comparison among the forests (Zheng et al., 2015; Zhu et al., 2013a). High fertilization rates can remove all possible N and P constraints in both young and old-growth forests (Cleveland and Townsend, 2006). In addition, plot size and fertilizer level in our forests were also the same as those in Costa Rica by Cleveland and Townsend (2006)." (Please also see Page 7 Lines 8-15 in the revised manuscript).

Based on our explanation above, if we use the slightly more realistic rates of N and P (~50 kg ha$^{-1}$ yr$^{-1}$), we guess that the low fertilization rates may be insufficient to affect soil N$_2$O emission in the old-growth forest, but it may have the same effects as our present study using the high rates in the two younger forests. Future studies will be carried out to test this case.

Reference:
Cleveland, C. C., and Townsend, A. R.: Nutrient additions to a tropical rain forest drive substantial soil carbon dioxide losses to the atmosphere, P. Natl. Acad. Sci. USA, 103, 10316-10321, 2006.
Liu, L., Gundersen, P., Zhang, T., and Mo, J. M.: Effects of phosphorus addition on soil microbial biomass and community composition in three forest types in tropical China, Soil Biol. Biochem., 44, 31-38, 2012.
Mo, J., Zhang, W., Zhu, W., Gundersen, P., Fang, Y., Li, D., and Wang, H.: Nitrogen addition reduces soil respiration in a mature tropical forest in southern China, Global Change Biol., 14, 403-412, 2008.
Zhang, W., Mo, J., Zhou, G., Gundersen, P., Fang, Y., Lu, X., Zhang, T., and Dong, S.: Methane uptake responses to nitrogen deposition in three tropical forests in

southern China, J. Geophys. Res, 113, 2008.

Zhu, F. F., Yoh, M., Gilliam, F. S., Lu, X. K., and Mo, J. M.: Nutrient limitation in three lowland tropical forests in southern China receiving high nitrogen deposition: insights from fine root responses to nutrient additions, PLoS One, 8, e82661, 2013.

Zheng, M., Huang, J., Chen, H., Wang, H., and Mo, J.: Responses of soil acid phosphatase and beta-glucosidase to nitrogen and phosphorus addition in two subtropical forests in southern China, Eur. J. Soil Biol., 68, 77-84, 2015.

Technical comments:

1) P7 line 1-5: you need to include a bit more detail on the chamber design: dimensions of the baseframe and lid (or chamber). Did you use a stiring fan, pressure valve? How did you seal the chamber to the lid?

**Answer:** Thank you for this good suggestion, and we have added this information in the text: "Each static chamber consisted of an anchor ring and a removable cover chamber. The anchor ring was a PVC pipe (25 cm diameter and 16 cm height) permanently anchored into the soil to 8 cm depth. During gas collection, a removable cover chamber (25 cm diameter and 30 cm height) was attached tightly to the anchor ring using a rubber O-ring seal." (Please also see Page 7 Line 24-25 and Page 8 Lines 1-2 in the revised manuscript).

In this study, we did not use a stiring fan, but we used the syringes to flush chamber gas three times to mix the headspace before each sampling. So, we have added this information: "Before each sampling, syringes were flushed three times with chamber gas to mix the headspace." (Please also see Page 8 Lines 5-6 in the revised manuscript).

2) P7 line 5: Change sentence to: '…and analyzed within 12 h on the gas chromatograph (Agilent 4890D) fitted…' (replaced 'in' with 'on').

**Answer:** Thank you, and we have replaced "in" with "on" in this sentence. (Please also see Page 8 Line 6 in the revised manuscript).

3) P7 line 10: 'The calculation of $N_2O$ fluxes followed the method of Holland et al. (1999), based on linear regression of' chamber gas concentration across time (changed 'across' to 'with').

**Answer:** Thank you. We have replaced "across" with "with" in this sentence. (Please also see Page 8 Line 18 in the revised manuscript).

4) P7 Line 11: was the soil temperature measured inside the chamber?

**Answer:** Yes, both soil temperature and soil moisture were measured inside the chamber. To make this clear, we have added this information in the text: "soil temperature (at 5 cm depth) and moisture (0−10 cm depth) inside each chamber, were measured…." (Please also see Page 8 Line 20 in the revised manuscript).

5) P7: line 16: I am not certain that the very general particle density value of 2.65 g/cm3 is appropriate to be used for your forest soils? Would you not expect a different particle density in the OG forest compared to the mixed/pine forests?

**Answer:** Thank you, and we agreed with your comment. However, in our study, we cautiously used the particle density value of 2.65 g cm$^{-3}$ just as an assumption value. This value has been suggested to applied in mineral soils of forests (Linn et al., 1984), and has been widely used in many tropical forests (Koehler et al., 2009; Rowlings et al., 2012; Zhu et al., 2013) This value was also used in our old-growth forest in a previous study (Zhang et al., 2012).

It is possible that the value may be different between forest types (old-growth vs. younger forests). However, the case that the same value (2.65 g cm$^{-3}$) was used in different ages of forest soils can also be found in other forest studies (Riley et al., 1995; Werner et al., 2006). In addition, because we using the WFPS focus on the comparison between treatments rather than between forest types in this study, whether or not using different particle density values to calculate WFPS may be of minor importance.

To make it more clear to the readers, we have replaced "….2.65 is the density of soil particles (g cm$^{-3}$)" with "….2.65 g cm$^{-3}$ is the assumed particle density in mineral soil of forests (Linn et al., 1984). It is possible that the particle density value may be different between forest types (old-growth vs. younger forests), but we focused on the comparison between treatments in this study, so this case is of minor importance." (Please also see Page 8 Lines 24-25 and Page 9 Lines 1-2 in the revised manuscript).

Reference:

Koehler, B., Corre, M. D., Veldkamp, E., Wullaert, H., and Wright, S. J.: Immediate and long-term nitrogen oxide emissions from tropical forest soils exposed to elevated nitrogen input, Global Change Biol., 15, 2049-2066, 2009.

Linn, D., and Doran, J.: Effect of water-filled pore space on carbon dioxide and nitrous oxide production in tilled and nontilled soils, Soil Sci. Soc. Am. J., 48, 1267-1272, 1984.

Riley, R. H., and Vitousek, P. M.: Nutrient dynamics and nitrogen trace gas flux during ecosystem development in montane rain forest, Ecology, 292-304, 1995.

Rowlings, D., Grace, P., Kiese, R., and Weier, K.: Environmental factors controlling temporal and spatial variability in the soil-atmosphere exchange of $CO_2$, $CH_4$ and $N_2O$ from an Australian subtropical rainforest, Global Change Biol., 18, 726-738, 2012.

Werner, C., Zheng, X. H., Tang, J. W., Xie, B. H., Liu, C. Y., Kiese, R., and Butterbach-Bahl, K.: $N_2O$, $CH_4$ and $CO_2$ emissions from seasonal tropical rainforests and a rubber plantation in Southwest China, Plant Soil, 289, 335-353, 2006.

Zhang, T., Zhu, W., Mo, J., Liu, L., Dong, S., and Wang, X.: Increased phosphorus availability mitigates the inhibition of nitrogen deposition on $CH_4$ uptake in an old-growth tropical forest, southern China, Biogeosciences, 8, 2011.

Zhu, J., Mulder, J., Wu, L., Meng, X., Wang, Y., and Dörsch, P.: Spatial and temporal

variability of $N_2O$ emissions in a subtropical forest catchment in China, Biogeosciences, 10, 1309-1321, 2013.

6) P8 line 3: How was $NH_4$ extracted from the soil?
**Answer:** Thank you for pointing out this question, and we have added this information in the text: "….after extraction with potassium chloride solution". (Please also see Page 9 Line 16 in the revised manuscript).

7) P8: line 8 & 10: $NO_3^-N$. '-' should not be a superscript
**Answer:** Thank you for this careful review, and we have replaced "$NO_3^-N$" with "$NO_3^--N$" in the text. (Please also see Page 9 Line 20 and Line 22 in the revised manuscript)

8) P10 line 3: change to: 'mixed, and pine forests, respectively (Fig. 4), with being significantly higher ($P = 0.001$) in the old-growth forest.
**Answer:** Thank you, and we have followed your suggestion to change the sentence to "mixed, and pine forests, respectively (Fig. 4), with being significantly higher ($P = 0.001$) in the old-growth forest" in the text. (Please also see Page 11 Lines 17-18 in the revised manuscript)

9) Page 9 line 16 delete 'were' and line 21: delete 'was'.
**Answer:** Thank you for this suggestion. We have deleted "were" and "was" in the corresponding positions in the text.

10) Page 10: line 3: change to 'mixed, and pine forests, respectively (Fig. 4), with being significantly higher ($P = 0.001$) in the old-growth forest'.
**Answer:** Thank you, and we have followed your suggestion to change the sentence to "mixed, and pine forests, respectively (Fig. 4), with being significantly higher ($P = 0.001$) in the old-growth forest" in the text. (Please also see Page 11 Lines 17-18 in the revised manuscript)

11) P 11 section 4.1 first paragraph: you may like to add that the variability of the data available could be due to soil type and also variability in climate.
**Answer:** It is a good suggestion, and we have added this information in this section: "Taken together, these data suggest a high variation in $N_2O$ emission among different study regions, possibly due to the difference in soil types and/or climatic conditions." (Please also see Page 13 Lines 4-6 in the revised manuscript)

12) P11 line 23-24: is this the same forest as in your study? If this is the case, replace with: ...in this old-growth forest, investigated previously by Fang et al (2008).
**Answer:** Yes, it is the same forest as in our study. According to your suggestion, we have replaced the sentence "….in the old-growth forest (Fang et al., 2008)" with "…. in this old-growth forest, investigated previously by Fang et al (2008)". (Please also see Page 13 Line 17 in the revised manuscript)

13) P13 line 12 'In spring, forest soil was enriched with inorganic N (accumulated during non-growing seasons)'you need to say that the non growing season is due to the lack of rainfall. Also comment on the pulsing effect (wetting dry soil triggers $N_2O$ emissions and other gases.

**Answer:** Thank you for this good suggestion. (1) We have added: "accumulated during non-growing seasons mainly due to the lack of rainfall" (Please also see Page 14 Line 23 in the revised manuscript). (2) We have replaced the sentence "conditions that would increase microbial consumption of soil $NH_4^+$ and/or $NO_3^-$ (Davidson et al., 2000), and thus greatly increase $N_2O$ production (Davidson et al., 2000; Butterbach-Bahl et al., 2004; Werner et al., 2006)." with "conditions that would generate a pulsing effect, because wetting dry soil will trigger emissions of $N_2O$ and other nitrogenous gases by stimulating microbial consumption of soil $NH_4^+$ and/or $NO_3^-$ (Davidson et al., 2000; Butterbach-Bahl et al., 2004; Werner et al., 2006)." (Please also see Page 14 Lines 24-25 and Page 15 Lines 1-2 in the revised manuscript)

14) P14 line 22: change to 'allowing us to reject the hypothesis that P addition causes greater decrease in $N_2O$ emission'.
**Answer:** Thank you, and we have followed your suggestion to change the sentence to "allowing us to reject the hypothesis that P addition causes greater decrease in $N_2O$ emission…." in the text. (Please also see Page 16 Line 23 in the revised manuscript)

15) P15 line 1-2: Under laboratory conditions, Sundareshwar et al. (2003) found a negative response of sediment $N_2O$ emission to nitrate addition. This sentence should be moved to the nitrogen section 4.3
**Answer:** Thank you for this suggestion, but we would like to make an explanation for this sentence. "nitrate addition" is a typo made by mistake, and we intended to use the phase "phosphate addition". We have replaced "nitrate addition" with "phosphate addition" in this sentence, and thus, this sentence may be appropriate to remain in the section "4.4 Effects of P addition on $N_2O$ emission". (Please also see Page 17 Line 4 in the revised manuscript)

16) Fig 3 & 4 legend line 3: delete 'before analysis' Fig 5 legend change to…" in the three control plots of the study forest…;
**Answer:** Thank you for these suggestions. We have deleted "before analysis" in both Fig.3 and Fig. 4 legend in line 3. Because we have five control plots in each forest, we replaced "….in the study forests" with "….in the five control plots of the study forests" in Fig. 5 legend.

**Referee #5**

General comments
This study presents a field experiment studying the effects of nitrogen and phosphorus additions on soil $N_2O$ emissions from nutrient rich and nutrient poor tropical forests. In general, the paper is well written and is highly relevant, and it provides valuable new information about the combined and individual effects of N and P fertilization on soil $N_2O$ emissions. Based on three referees and the response of the authors to them, the authors have already addressed several issues related to e.g. the high N fertilization rates, effects of P on alleviation of $N_2O$ emissions, which has greatly improved the quality of the paper. However, I have few additional comments that are mainly related to the gas analysis of $N_2O$, presentation and interpretation of the results. I consider this work important and worth publishing after addressing the points below.
**Answer:** Thank you very much for these positive comments.

Specific comments
1) Page 2, lines 24-25: Could you explain how and in what respect the tropical forests have shown an increase in soil $N_2O$ emissions compared to temperate and boreal forest soils? Is this due to increased atmospheric N deposition, or do you refer merely to a pure comparison of the $N_2O$ emission rates from these ecosystems?
**Answer:** Thank you for this good suggestion. The reviewer suggested that we need to explain how and in what respect the tropical forests have shown an increase in soil $N_2O$ emissions compared to temperate and boreal forest soils, and also pointed out whether this may be due to increased atmospheric N deposition? −We agreed with this suggestion and have added this information in the text. Specifically, we have replaced this sentence: "Compared with temperate and boreal forests, tropical forests have shown a great increase in soil $N_2O$ emissions (Matson and Vitousek, 1990)" with "Because tropical forest soils are often rich in N but poor in P, they are less able to retain external N input (Hall and Matson, 1999). With the greatest increases of atmospheric N deposition occurred in tropical regions (Galloway et al., 2008), tropical forests have shown a great increase in soil $N_2O$ emissions, compared with temperate and boreal forests (Matson and Vitousek, 1990)." (Please also see Page 2 Line 25 and Page 3 Lines 1-4 in the revised manuscript)

Reference:
Galloway, J. N., Townsend, A. R., Erisman, J. W., Bekunda, M., Cai, Z., Freney, J. R., Martinelli, L. A., Seitzinger, S. P., and Sutton, M. A.: Transformation of the nitrogen cycle: recent trends, questions, and potential solutions, Science, 320, 889-892, 2008.
Hall, S. J., and Matson, P. A.: Nitrogen oxide emissions after nitrogen additions in tropical forests, Nature, 400, 152-155, 1999.
Matson, P. A., and Vitousek, P. M.: Ecosystem approach to a global nitrous oxide budget, Bioscience, 40, 667-671, 1990.

2) Page 3, lines 2-5: I consider that in addition to the mentioned factors, the poor knowledge in factors controlling $N_2O$ emissions in tropical forests is also due to the rather small number of studies from these ecosystems.

**Answer:** Thank you, and we agreed with your comment. Accordingly, we have added this information in the text: "This is not only because…, but also because only a small number of studies in tropical forests is available (Dalal and Allen, 2008; Liu and Greaver, 2009)." (Please also see Page 3 Lines 8-9 in the revised manuscript)

Reference:

Dalal, R. C., and Allen, D. E.: Greenhouse gas fluxes from natural ecosystems, Australian Journal of Botany, 56, 369-407, 2008.

Liu, L., and Greaver, T. L.: A review of nitrogen enrichment effects on three biogenic GHGs: the CO2 sink may be largely offset by stimulated N2O and CH4 emission, Ecol. Lett., 12, 1103-1117, 2009.

3) Page 3, lines 11-13: I would mention here also other losses of N, such as leaching losses, emissions of $N_2$, NO, $NH_3$, and HONO, which all are signs of N saturation.

**Answer:** Thank you, and we agreed with this comment. Following your suggestion, we have replaced this sentence: "….leading to rapid N losses via $N_2O$ emission." with "….leading to rapid N losses via liquid leaching and gases emission (such as $N_2$, $N_2O$, NO, $NH_3$, and HONO)." (Please also see Page 3 Lines 16-17 in the revised manuscript)

4) Page 5, lines 18-20: I would like to see here more description of where the N is retained (soil, above ground biomass, below ground biomass, microbial biomass), and in what forms are the N losses from the soil (leaching, gaseous losses, what gas species etc).

**Answer:** Thank you for these good suggestions.

First, the reviewer suggested that we need to add more description of where the N is retained (soil, above ground biomass, below ground biomass, microbial biomass). ──We have followed this suggestion and replaced "….net retention of 22−28 kg N $ha^{-1}yr^{-1}$ in the two younger forests…." with "….22−28 kg N $ha^{-1}yr^{-1}$ were retained in the upper 20cm soil and the plant biomass (including canopy trees, understory plants and forest litter) in the two younger forests," (Please also see Page 6 Lines 6-7 in the revised manuscript)

Second, the reviewer also suggested that we need to add the forms of N losses from the soil (leaching, gaseous losses, what gas species etc). −We have also followed this suggestion and replaced "….net loss of 8−16 kg N $ha^{-1}yr^{-1}$ from the soil in the old-growth forest (Fang et al., 2008)." with "and that a net loss of 8−16 kg N $ha^{-1}yr^{-1}$ mainly via dissolve inorganic N ($NH_4^+$ and $NO_3^-$) leaching and soil $N_2O$ emission occurred in the old-growth forest (Fang et al., 2008)." (Please also see Page 6 Lines 7-9 in the revised manuscript)

Reference:

Fang, Y. T., Gundersen, P., Mo, J. M., and Zhu, W. X.: Input and output of dissolved organic and inorganic nitrogen in subtropical forests of South China under high air pollution, Biogeosciences, 5, 339-352, 2008.

5) Page 7, lines 5-7: could you give more details of the gas chromatographic analysis. I'm missing information of the used columns, oven temperature, flow rates, carrier and make-up gases. Especially, I'm interested and slightly concerned whether $CO_2$ was allowed to enter the ECD, or whether it was trapped chemically (e.g. ascarite) as if $N_2$ is used as a carrier gas, and CO2 is allowed to enter the ECD, this may bias the $N_2O$ analysis and lead to overestimated $N_2O$ fluxes as described by Zheng et al. (2008). Zheng X., et al., 2008. Quantification of $N_2O$ fluxes from soil-plant systems may be biased by the applied gas chromatograph methodology. Plant and Soil, 311: 211-234.

**Answer:** Thank you for this good suggestion. Following your suggestion, we have added this information in the text: "Two stainless steel columns (pre-column and main-column was 1m and 3m in length, respectively) packed with Porapak Q were used to separate $N_2O$. The oven temperature and ECD temperature was 55 ºC and 330 ºC, respectively. To avoid the interference of $CO_2$ from the gas samples which can lead to overestimation of $N_2O$ fluxes as suggested by Zheng et al. (2008), we used $N_2$ as the carrier gas (flow rate of 35mL $min^{-1}$) and introduced 10% of $CO_2$ in $N_2$ as the make-up gas (flow rate of 2mL $min^{-1}$) into the ECD (Wang et al., 2010). Through introducing high concentration and low flow rate of $CO_2$ into the ECD, the interference of $CO_2$ from the gas samples is negligible (Wang et al., 2010)." (Please also see Page 8 Lines 7-13 in the revised manuscript)

Reference:

Wang, Y., Wang, Y., and Ling, H.: A new carrier gas type for accurate measurement of $N_2O$ by GC-ECD, Adv. Atmos. Sci., 27, 1322-1330, 2010.

Zheng, X., Mei, B., Wang, Y., Xie, B., Wang, Y., Dong, H., Xu, H., Chen, G., Cai, Z., and Yue, J.: Quantification of $N_2O$ fluxes from soil–plant systems may be biased by the applied gas chromatograph methodology, Plant Soil, 311, 211-234, 2008.

6) Page 8, lines 13-19: I'm missing information whether you tested the data for normality and equality of variances. Naturally, if these criteria were met, the use of parametric tests are justified, otherwise non-parametric tests should be used. Please, clarify this.

**Answer:** Thank you for this comment. In fact, the data in our study have been tested for normality using the Kolmogorov-Smirnov test and for equality of variance using Levene's test. Those data that did not meet the requirements of normality and equality of variance have been log-transformed before statistical analysis. Following your suggestion, we have added this information in the text: "Data were tested for normality (Kolmogorov-Smirnov test) and equality (Levene's test) of variances, and were log-transformed for analysis if they did not meet the requirements of normality

or equality of variances." (Please also see Page 10 Lines 4-6 in the revised manuscript)

7) Page 12, lines 13-14: This line is almost identical to the sentence from page 8, lines 23-24. Please, modify.
**Answer:** Thank you for this careful review, and we agreed with this comment. Accordingly, we have modify this sentence "Soil temperature in all plots in the three forests showed a similar seasonal pattern, increasing from spring to summer and decreasing from fall to winter (Fig. 1)" to "Overall, soil temperature increased from spring to summer but decreased from fall to winter in all the forest plots (Fig. 1)." without changing its initial meaning. (Please also see Page 14 Lines 7-8 in the revised manuscript)

8) Page 12, lines 16-18: was the difference in mean soil temperature statistically significant between the three forests? If yes, please give the p-value. Also, were the $N_2O$ emission rates across different forests significantly different? If yes, please give the p-value here. In other words, if the above mentioned differences were not statistically significant, you cannot claim that soil temperature does not explain the $N_2O$ emission pattern across the forests.
**Answer:** Thank you, and we have followed your suggestions.

First, the reviewer pointed out whether the difference in mean soil temperature was statistically significant between the three forests? —— Yes. Repeated measures ANOVA showed the significant differences ($P < 0.001$) in mean soil temperatures between each forest (as we have mentioned in the Result section, Page10 Lines 15-16). Thus, to make it clear to the readers, we have added "(statistical difference of $P < 0.001$ between each forest)" here (Please also see Page 14 Line 11 in the revised manuscript).

Second, the reviewer pointed out whether the difference in the $N_2O$ emission rates was significantly different across different forests? —— Yes. Mean $N_2O$ emission rate was significantly higher in the old-growth forest than in the mixed ($P = 0.001$) and pine ($P = 0.005$) forests (as we have mentioned in the Result section, Page11 Lines 17-19). To make it clear to the readers, we have also added "(with being significantly higher in the old-growth forest than in the mixed ($P = 0.001$) and pine ($P = 0.005$) forests; Fig. 4)." here (Please also see Page 14 Lines 12-13 in the revised manuscript).

Third, the reviewer pointed out if the above mentioned differences were not statistically significant, it is incorrect to claim that soil temperature does not explain the $N_2O$ emission pattern across the forests. —— Thank you for this comment. Both mean soil temperature and mean $N_2O$ emission rates are statistically significant across forests (as we mentioned above), and thus, this suggests "a limited ability of soil temperature to explain the pattern in $N_2O$ emission across forests". (Please also see Page 14 Lines 13-14 in the revised manuscript).

9) Page 12-13, chapter 4.2: You present simple correlation analysis of $N_2O$ emissions

against soil temperature or soil moisture, and use robust linear regression to explain the N$_2$O emissions (Fig. 5). Based on the scatter plots, it seems that there is an exponential relationship between at least N$_2$O fluxes and soil temperature. Did you try to fit also non-linear models to the data? Also, as the correlation between both N$_2$O flux and soil temperature, and N$_2$O flux and soil moisture are highly significant, did you try to build a regression model including both soil temperature and soil moisture as parameters? This might be worth the effort.

**Answer:** Thank you for these comments and suggestions.

First, the reviewer pointed out that an exponential relationship may occur at least between N$_2$O fluxes and soil temperature, based on the scatter plots, and asked whether we have tried to fit also non-linear models to the data? —— Thank you for this comment. In fact, we have used some suitable exponential regression models to build the relationships between N$_2$O flux and soil WFPS, or between N$_2$O flux and soil temperature, but the coefficients of determination (R$^2$) of the exponential regression models were lower than those of the linear regression model used in this study. For example, we chose some suitable exponential regression models (i.e. y=a×exp(b×x), y=a×exp(x/b)+c, y=a×exp(x×b)+c), and the R$^2$ of the models are 0.102−0.110 between N$_2$O and soil WFPS, 0.166−0.167 between N$_2$O and soil temperature in the old-growth forest; 0.176−0.180 between N$_2$O and soil WFPS, and 0.098−0.099 between N$_2$O and soil temperature in the mixed forest; 0.225−0.226 between N$_2$O and soil WFPS, and 0.071−0.082 between N$_2$O and soil temperature in the pine forest. All these coefficients were lower than those of the linear regression model used in this study (Please see Table 4). In addition, some other non-linear models (such as power regression models) were also tried, but the coefficients of determination were also lower than those of the linear regression regression. Therefore, based on above reasons, we used the linear regression model in this study.

Second, the reviewer suggested a regression model including both soil temperature and soil moisture as parameters.—— Thank you for this good suggestion, and we have followed it. Because liner regression model had the better fitting effect in our study as we mentioned above, we added the liner regression models including both soil temperature and soil moisture as parameters in "Table 4". Based on this added model, we have also added more information in the text: "In the control plots, soil temperature and WFPS showed a significant positive linear relationship with soil N$_2$O emission (Fig. 5), and explained 9−17% and 12−23% of N$_2$O fluxes variation across the forests (Table 4). The models that included soil temperature and WFPS as parameters showed the higher R$^2$ values (22−28%; Table 4)" (Please also see Page 11 Lines 19-22 in the revised manuscript), and also added "Compared to the models with soil temperature and N$_2$O fluxes as parameters, the R$^2$ values of the models with soil WFPS and N$_2$O fluxes as parameters were not much higher (Table 4). However, mean soil WFPS showed comparable dynamics to mean N$_2$O emission, with the highest in the old-growth forest and lowest in the pine forest (Fig. 2)" in the Discussion section (Please also see Page 14 Lines 16-19 in the revised manuscript). The added information above helps us further improve the manuscript.

10) Page 13, lines 1-7: Based on only two soil sampling occasions (Feb 2007 and Aug 2009) it is very uncertain to conclude how the soil inorganic N concentrations developed during the different seasons. For instance, a soil sampling in February 2007 does not support that the soil was enriched with inorganic N, and also a soil sampling in August 2009 does not support that the inorganic N had decreased during the growing season, as there were no measurements during the growing season. Please, discuss these uncertainties, and if possible bring in material and references to support your conclusions.

**Answer:** Thank you very much for these constructive comments and suggestions, and we agreed with your comments. Although seasonal variances of soil inorganic N concentrations were not measured in this study, they were measured by our previous study in the same forests (Mo et al., 2003). Using ion exchange resin method, our previous study found that soil inorganic N concentrations ($NH_4^+$ plus $NO_3^-$) showed significant seasonal variations in the three forests, with the following order: spring (total mean value: 47.64±14.67 μg per day $g^{-1}$ dry resin) > fall (23.51±2.30 μg per day $g^{-1}$ dry resin) > winter (18.76±2.06 μg per day $g^{-1}$ dry resin) > summer (16.81±3.29 μg per day $g^{-1}$ dry resin) (Mo et al., 2003). Thus, this pattern supported our discussion in the text: "In spring, forest soil was enriched with inorganic N…." (Page 14 Lines 22-23 in the revised manuscript) and "In fall and winter, both the lower soil inorganic N (decreased after growing seasons)…." (Page 15 Lines 3-4 in the revised manuscript).

Accordingly, following your suggestion, we have added this reference (Mo et al., 2003) to support our conclusion in the text: "In spring, forest soil was enriched with inorganic N…. (Mo et al., 2003)" (Please also see Page 14 Lines 22-24 in the revised manuscript) and "In fall and winter, both the lower soil inorganic N (decreased after growing seasons) (Mo et al., 2003)" (Please also see Page 15 Lines 3-4 in the revised manuscript).

Reference:
Mo, J. M., Brown, S., Peng, S. L., and Kong, G. H.: Nitrogen availability in disturbed, rehabilitated and mature forests of tropical China, For. Ecol. Manage., 175, 573-583, 2003.

11) Table 2 and e.g. page 16, lines 18-20: The values in soil pH, inorganic N, organic C, microbial biomass and P in Table 2 are only from one sampling occasion, approximately two years from the start of the experiment. Also, the comparison between the fertilization treatments is conducted with data from one time sampling only, while the fertilization was conducted every second week over a two-year period. I see here a problem when comparing the effects of the fertilization. Firstly, I think it would be best to compare the soil N (and other measures) status before and after the treatments. But in this comparison, the timing of the sampling is important as the soil N (and other) have strong seasonality, which may be larger than the treatment effect. As the soil sampling before the experiment was in the spring (February 2007), and the soil sampling after the experiment was during summer (August 2009), it is very

difficult to know whether the differences result from the treatments or the seasonal variation in soil N. My other concern is that the different plots may have differed between each other already before the experiment. Did you test this? Overall, I think it is very difficult to conclude that the fertilization did or did not influence the soil N status in the experiment. Please, discuss these uncertainties or be more careful in interpreting the results, unless there is more data to support these findings.

**Answer:** Thank you very much for these constructive comments, and we would like to response to these comments point by point below.

First, the reviewer suggested that it would be best to compare the soil N (and other measures) status before and after the treatments, and that the timing of the sampling is also important as the soil N (and other) have strong seasonality, which may be larger than the treatment effect. —— Thank you for this suggestion. Firstly, it is a good method to compare the soil variables before and after the treatments, as suggested by the reviewer, but this method may not be suitable for our present study using long-term and on-going fertilization treatments, because it may be a little difficult to evaluate the difference caused by treatments or seasonality if we compared the results from different sampling periods after the treatments with those from before the treatments (Please note that we only measured soil properties once in February 2007 before fertilization). However, in our study, we have set up the control plots in all the three forests, which allow us to know about the treatment effects by the comparison between the fertilization plots and the control plots in the same sampling period. This method of studying treatment effects by setting up control plots has also been widely used in many forest studies (Treseder et al., 2001; Cleveland and Townsend et al., 2003; Hall and Matson, 2003; Davidson et al., 2008; Koehler et al, 2009; etc). Accordingly, we hope that our method of comparing the treatment plots with those in the control plots is also feasible. Secondly, we agreed that the timing of the sampling is also important because the soil variables may have seasonality, as suggested by the reviewer. For this reason, we now have showed all the soil properties values measured during our study period (in August 2007, February 2008, August 2008, February 2009, and August 2009), rather than one sampling occasion (Please also see Table 2 in the revised manuscript, and Table S2-S4 in the supporting information). We measured soil properties in February and August, mainly because (1) February and August is within the dry and wet season, respectively, in our study region, and (2) our study region had typical seasonal pattern, with the wettest and warmest during wet season and the driest and coldest during dry season. (Please also see Page 6 Lines 18-22 in the revised manuscript).

Second, the reviewer pointed out that the soil sampling before the experiment was in the spring (February 2007), and the soil sampling after the experiment was during summer (August 2009), so it is very difficult to know whether the differences result from the treatments or the seasonal variation in soil N. ——Thank you very much for this constructive comment. Firstly, we now have showed all the soil sampling data (August 2007, February 2008, August 2008, February 2009, and August 2009) rather than one time of soil sampling data (August 2009). Secondly, we now have analyzed soil variables using the method of repeated measures ANOVA, and this statistical

analyses method could evaluate the treatment differences based on different sampling periods. (Please also see Table 2 in the revised manuscript, and Table S2-S4 in the supporting information). Therefore, we hope that the above improvements could help us evaluate the treatment differences properly.

Third, the reviewer asked whether the different plots may have differed between each other already before the experiment. ——Thank you for this constructive comment. In fact, we have measured the soil properties in all the plots before the treatments, and we found no statistical difference of soil properties among the plots in each forest (Please also see Table S1 in the supporting information).

Fourth, the reviewer pointed out that it is difficult to conclude that the fertilization did or did not influence the soil N status in the experiment, and suggested us to discuss these uncertainties or be more careful in interpreting the results, unless there is more data to support these findings. ——Thank you very much for this comment. We have added more soil properties data to support our findings (Table 2, Table S2, Table S3 and Table S4) and used the repeated measures ANOVA to analyze the treatment effects rather than one-way ANOVA, which could rule out the interference of sampling times. Thus, we hope that these improvements will allow us to draw the conclusions more credibly.

Fifth, because we added more data of soil properties in the revised manuscript, we have made the revision on the description of the Results section (3.3 soil properties) **from** "Soil pH did not change after addition of fertilizers in the old-growth and pine forests, but significantly decreased after NP-addition in the mixed forest (Table 2). Soil $NH_4^+$ concentrations significantly increased after P- and NP-addition in the old-growth forest, while NP-addition significantly decreased soil $NO_3^-$ and $NH_4^+$ concentrations in the old-growth and pine forests, respectively. N-addition significantly decreased soil total inorganic N ($NH_4^+ + NO_3^-$) concentrations in the pine forest. No treatment effect occurred on soil organic C in the old-growth and pine forests, while both P- and NP-addition significantly increased soil organic C in the mixed forest. Soil microbial biomass C significantly increased after NP-addition in the old-growth forest and after N-, P- and NP-addition in the mixed forest. Although not always statistically significant, both P- and NP-addition increased soil available P concentrations in all the forests compared to the control plots" **to** "Repeated measures ANOVA showed that soil pH significantly increased after P-addition in the old-growth forest (Table 2). Soil $NO_3^-$ concentrations significantly decreased after P-addition in the old-growth and mixed forests, and significantly increased after N-addition in the pine forest. Soil $NH_4^+$ concentrations and total inorganic N ($NH_4^+ + NO_3^-$) concentrations had no response to either N- or P-addition in any forest. Soil available P concentrations significantly increased after P-addition in all the forests. Soil organic C significantly increased after N-addition in the mixed and pine forests, but not in the old-growth forest. Soil microbial biomass C significantly increased after P-addition in the old-growth forest and after N-addition in the mixed forest. Interaction of combined N and P additions occurred in soil AP concentrations and microbial biomass C in the old-growth forest, and in soil pH and $NO_3^-$ concentrations in the mixed forest" (Please also see Page 11 Lines 5-13 in the revised manuscript).

Although we added more data of soil properties, those results of soil properties supporting our findings did not change, that is (1) "In the old-growth forest, we found no increase in soil organic C, microbial biomass C (Table 2),…" (Please also see Page 15 Lines 19-20 in the revised manuscript); (2) "As a result, no significant increase in soil inorganic N ($NH_4^+$ and $NO_3^-$) was observed after N addition in the old-growth forest (Table 2)." (Please also see Page 15 Line 25 and Page 16 Lines 1-2 in the revised manuscript); (3) "despite no significant increase in soil total inorganic N following N addition, a significant increase in soil microbial biomass C and soil organic C was observed in the mixed forest, as well as a significant increase in soil organic C in the pine forest (Table 2)." (Please also see Page 16 Lines 7-9 in the revised manuscript); (4) "NP addition did not significantly affect soil total inorganic N ($NH_4^+$ plus $NO_3^-$) (Table S2)." (Please also see Page 19 Lines 1-2 in the revised manuscript). However, only the sentence of "However, we found no significant change in soil total inorganic N ($NH_4^+$ plus $NO_3^-$) after approximately 2 years of P addition in all forests, despite a significant increase in $NH_4^+$ in the old-growth forest (Table 2)." should be replaced with "However, we found no significant change in soil total inorganic N ($NH_4^+$ plus $NO_3^-$) after P addition in all forests, despite a significant decrease in $NO_3^-$ in the old-growth and mixed forests (Table 2)." (Please also see Page 17 Lines 16-18 in the revised manuscript), but the revision of this sentence did not affect our main finding in the Discussion.

Again, we appreciated the reviewer for the above comments and suggestions.

Reference:

Cleveland, C. C., and Townsend, A. R.: Nutrient additions to a tropical rain forest drive substantial soil carbon dioxide losses to the atmosphere, P. Natl. Acad. Sci. USA, 103, 10316-10321, 2006.

Davidson, E. A., Nepstad, D. C., Ishida, F. Y., and Brando, P. M.: Effects of an experimental drought and recovery on soil emissions of carbon dioxide, methane, nitrous oxide, and nitric oxide in a moist tropical forest, Global Change Biology, 14, 2582-2590, 2008.

Hall, S. J., and Matson, P. A.: Nutrient status of tropical rain forests influences soil N dynamics after N additions, Ecol. Monogr., 73, 107-129, 2003.

Koehler, B., Corre, M. D., Veldkamp, E., Wullaert, H., and Wright, S. J.: Immediate and long-term nitrogen oxide emissions from tropical forest soils exposed to elevated nitrogen input, Global Change Biol., 15, 2049-2066, 2009.

Treseder, K. K., and Vitousek, P. M.: Effects of soil nutrient availability on investment in acquisition of N and P in Hawaiian rain forests, Ecology, 82, 946-954, 2001.

12) Fig. 5: Is this data from the control plots only? Please, specify which data was used.

**Answer:** Yes. The data were from the control plots. To make it clear to the readers, we have added "….in five control plots of the study forests" in the figure legend. (Please also see the legend of Fig. 5)

Technical corrections

13) Page 2, line 12: add "atmospheric lifetime" inside the parenthesis.

**Answer:** Thank you for this suggestion, and we have added "atmospheric lifetime" inside the parenthesis. (Please also see Page 2 Line 13 in the revised manuscript)

14) Page 4, line 15, and line 19: change a N-rich to "an N-rich"

**Answer:** Thank you for this careful review. Following your suggestion, we have replaced "a N-rich" to "an N-rich". (Please also see Page 4 Line 21 and Page 5 Line 7 in the revised manuscript)

15) Page 6, line 7: I assume that you mean wet N deposition. If so, please add the word "wet" to the "Inorganic N deposition…". Or if this is a sum of wet and dry deposition, please clarify it.

**Answer:** Thank you for this suggestion. We mean wet N deposition in this sentence, so we have added "wet" to the "Inorganic N deposition…" in the text. (Please also see Page 6 Line 22 in the revised manuscript)

16) Page 8, line 25; page 9, line 9; page 10, line 2 and elsewhere in the paper: I would harmonize the use of decimal places, preferably round them to one decimal place. At least with $N_2O$ fluxes, I don't think the precision of the measurement is high enough to give the emissions with the accuracy of two decimal places.

**Answer:** Thank you for this suggestion. Following your suggestion, we have harmonized the use of decimal places, from two decimal places to one decimal place, throughout the text. (Please also see Page 2 Lines 2-3; Page 10 Line 14, 24; Page 11 Line 17; Page 12 Lines 1-7; Table 1 and Table 2 in the revised manuscript)

17) Page 12, line 22: add "WFPS" and "the" to the sentence: "highest WFPS in the old-growth forest and the lowest in the pine forest"

**Answer:** Thank you, and we have followed this suggestion to add "WFPS" and "the" to the sentence, and this sentence is now "the highest WFPS in the old-growth forest and the lowest WFPS in the pine forest". (Please also see Page 14 Line 18-19 in the revised manuscript)

18) Tables 1 and 2. Please, give the numbers with one decimal place.

**Answer:** Thank you. We have followed this suggestion to give the numbers with one decimal place in the two tables. (Please also see "Table 1" and "Table 2" in the revised manuscript)

*Acknowledgement*

*We appreciate four referees for the above comments and suggestions which help us improve the manuscript greatly.*